



# Wind tunnel experiments on wind turbine wakes in yaw: Effects of inflow turbulence and shear

Jan Bartl[1], Franz Mühle[2], Jannik Schottler[3], Lars Sætran[1], Joachim Peinke[3,4], Muyiwa Adaramola[2], and Michael Hölling[3]

[1] Department of Energy and Process Engineering, Norwegian University of Science And Technology, Trondheim, Norway
[2] Faculty of Environmental Sciences and Natural Resource Management, Norwegian University of Life Sciences, Ås, Norway
[3] ForWind, Institute of Physics, University of Oldenburg, Oldenburg, Germany
[4] Fraunhofer IWES, Oldenburg, Germany

*Correspondence to:* Jan Bartl (jan.bartl@ntnu.no)

**Abstract.** The wake characteristics behind a yawed model wind turbine exposed to different customized inflow conditions are investigated. Laser Doppler Anemometry is used to measure the wake flow in two planes at $x/D$=3 and $x/D$=6 while the turbine yaw angle is varied from $\gamma = [-30°, 0°, +30°]$. The objective is to assess the influence of grid-generated inflow turbulence and shear on the mean and turbulent flow components.

The wake flow is observed to be asymmetric with respect to negative and positive yaw angles. A counter-rotating vortex pair is detected creating a kidney-shaped velocity deficit for all inflow conditions. Exposing the rotor to non-uniform shear inflow changes the mean and turbulent wake characteristics only insignificantly. At low inflow turbulence the curled wake shape and wake center deflection are more pronounced than at high inflow turbulence. For a yawed turbine the rotor-generated turbulence profiles peak in regions of strong mean velocity gradients, while the levels of peak turbulence decrease at approximately the same rate as the rotor thrust.

## 1 Introduction

In the light of a steadily increasing worldwide use of wind energy, optimized control for wind farms has become a focus area of research. The reduced wind speeds in the wake leave significantly less energy for downstream turbines causing wind farm power losses up to 20% (Barthelmie et al., 2010). At the same time increased turbulence levels in the wake lead to higher fatigue loads on downstream rotors, which experience an increased probability for component failure (Thomsen and Sørensen, 1999). In order to mitigate these unfavorable consequences of wake impingement, different wind farm control methods have been suggested for optimizing the total power output and minimizing loads on a wind farm's individual turbines (Knudsen et al., 2014; Gebraad et al., 2015).

These methods include the reduction of the upstream turbine's axial-induction by varying its torque or blade pitch angle (Annoni et al., 2016; Bartl and Sætran, 2016) as well as wake redirection techniques, which intentionally apply an uneven load distribution on the front row rotors. In Fleming et al. (2015) different wake deflection mechanisms have been discussed with respect to higher wind farm power production and rotor loads. As individual pitch control has been shown to cause high struc-



tural loads and current turbine designs do not feature a degree of freedom in tilt direction, yaw actuation has been concluded to be a very promising technique.

For the development of wake deflection strategies by yaw misalignment, the characteristics of the mean and turbulent wake flow behind a yawed turbine have to be understood in detail. Besides the turbine's geometry and operational state, the wake
flow is strongly dependent on the atmospheric conditions which represent the inflow state to the turbine. The stability of the atmospheric boundary layer can be described by height-dependent distributions of potential temperature, wind direction (veer), velocity distribution (shear) and turbulence intensity (Vollmer et al., 2016). As it is rather impossible to simulate realistic atmospheric conditions in a wind tunnel environment, these parameters have to be investigated separately Therefore, the present study investigates the dependency of the wake flow behind yawed turbines for different customized inflow conditions. The wind
tunnel study intends to shed light on the effects of non-uniform shear and inflow turbulence levels on the wake characteristics. Wind tunnel wake experiments have the advantage of being conducted in controlled laboratory environment. Thus, intentional variations of inflow conditions and turbine operating points can help to gain a deeper understanding of the effects on the wake flow. They furthermore can serve as validation data of numerical results and a base for the fine-tuning of engineering wake models.

An early set of experimental studies on the wake of a yawed turbine was reported by Grant et al. (1997), in which they used optical methods in the wake behind a model turbine of $D$=0.90 m to track the tip vortices and calculate wake deflection and expansion. In a follow-up study, Grant and Parkin (2000) presented phase-locked particle image velocimetry (PIV) measurements in the wake. The measured circulation in the wake showed clear asymmetries for positive and negative yaw angles. An asymmetric wake was also reported by Haans et al. (2005), who found non-symmetric tip vortex locations behind a yawed
model turbine of $D$=1.20 m. Another yaw experiment was conducted by Medici and Alfredsson (2006) on a small model turbine of $D$=0.12 m. They reported a clear cross-stream flow component deflecting the wake laterally. These experimental results were later used by Jiménez et al. (2010) as verification data for a wake deflection model for yawed turbines. Based on large eddy simulations (LES) around a yawed actuator disc they developed a simple analytical model that is able to predict the wake skew angle and wake velocity deficit in the far wake. An engineering model for the axial induced velocity on a yawed turbine
was developed by Schepers (1999), which was based on inflow measurements in front of different yawed turbines.

An extensive study of flow and load characteristics on a yawed wind turbine rotor on a $D$=4.50 m rotor was presented by Schepers et al. (2014). In the so-called Mexnext project, a comparison of twenty different computations with detailed PIV and load measurements revealed modeling deficiencies while simultaneously shedding light on complex instationary flow at the rotor. The topic of utilizing yaw misalignment for improved wind farm control was thoroughly investigated by Fleming et al.
(2015) and Gebraad et al. (2016). They analyzed wake mitigation strategies by using both a parametric wake model and the advanced computational fluid dynamics (CFD) tool SOWFA. In another LES investigation Vollmer et al. (2016) studied the influence of three atmospheric stability classes on the wake characteristics behind a yawed turbine rotor. A strong dependency of the wake shape and deflection on the stability is found, showing significantly higher wake deflection for a stable atmosphere than for neutral or convective conditions. Another LES study on yaw misalignment was performed by Wang et al. (2017),
who highlighted the importance of including nacelle and tower structures in the computational model when comparing with





experimental results.

Yaw angle dependent turbine performance and near-wake measurements were performed by Krogstad and Adaramola (2012). They found a power decrease proportional to $cos^3(\gamma)$ and showed that the near-wake deflection is dependent on the turbine's tip speed ratio. A combined experimental and computational wake study for a larger range of donstream distances was recently

reported by Howland et al. (2016). The wake behind a yawed small drag disc of $D$=0.03 m was analyzed, describing the formation of a curled wake shape by a counter-rotating vortex pair. An extensive contribution to the field of yawed turbine wakes was recently made by Bastankhah and Porté-Agel (2016). In an experimental PIV study on a model turbine of D=0.15 m an asymmetric flow entrainment in the wake by both mean and turbulent momentum fluxes was shown. Moreover, an analytical model for the far wake of a yawed turbine was developed based on self-similar velocity and skew angle distributions.

An experimental study on the interaction of two model wind turbines was conducted by Schottler et al. (2016) showing clear asymmetries of the downstream turbine power output with respect to the upstream turbine's positive or negative yaw angle. In a follow-up study the asymmetry was ascribed to a strong shear in the inflow, which caused an asymmetry in the opposite direction when the sheared inflow was vertically inverted (Schottler et al., 2017b). These studies encouraged a more detailed investigation of the inflow-dependent wake flow behind a yawed turbine. As for the present study, we aim to close the gap

between turbine interactions for yaw-controlled wind farms by presenting high-fidelity wake measurement data at controlled inflow conditions. The influence of turbulence and shear in the inflow on the wake's shape, deflection and symmetry with respect to yaw angle is quantified. This work is part of a joint experimental campaign by the NTNU Trondheim and ForWind in Oldenburg. A second paper by Schottler et al. (2017a) compares the wakes of two different model wind turbines, adding two-point statistics to the evaluation.

## 2   Experimental setup

### 2.1   Turbine model, inflow & operating conditions

**Turbine model**

The wind turbine model used for this study has a rotor diameter of $D$=0.90 m with a hub diameter of $D_{hub}$=0.090 m. The tower and nacelle structure of the turbine is a slimmer re-design of the turbines used in previously used in Bartl and Sætran (2017).

The tower thickness and the nacelle length have been significantly reduced in size in order to minimize their impact on the wake flow behind the yawed rotor. Photographs of the turbine exposed to different inflow conditions are shown in Figure 1. The blades are milled in aluminum and based on a NREL S826 airfoil, which was originally designed at the National Renewable Energy Laboratory (NREL). The rotor turns in counter-clockwise direction when observed from an upstream point of view. The rotation is controlled via an electric servo motor of the type 400W Panasonic LIQI, which is located inside the nacelle.

The frequency-controlled motor ensures a rotation at constant rotational speed, while the excessive power is burned off in an external resistor. The blade pitch angle was fixed to $\beta = 0°$ for the entire experiment.





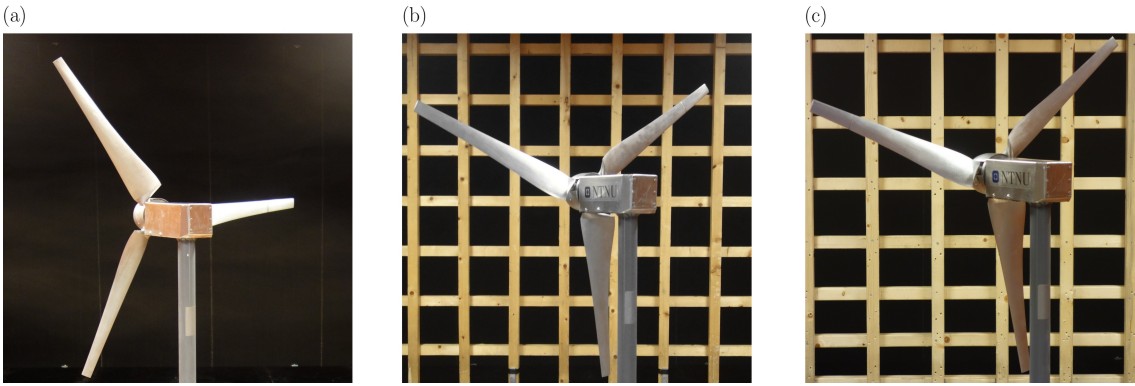

**Figure 1.** Yawed model wind turbine exposed to different inflow conditions: **(a)** $TI$=0.23%, uniform **(b)** $TI$=10.0%, uniform **(c)** $TI$=10.0%, non-uniform shear.

**Table 1.** Characteristics of the three different investigated inflow conditions

| Inflow | TI [%] | spatial uniformity | power law coeff. $\alpha$ |
|--------|--------|--------------------|---------------------------|
| A | 0.23 | uniform | 0 |
| B | 10.0 | uniform | 0 |
| C | 10.0 | non-uniform | 0.11 |

**Scaling and blockage**

The experiments were performed at the low-speed wind tunnel at the Norwegian University of Science and Technology (NTNU) in Trondheim, Norway. The test section is $11.15\,\mathrm{m}$ long with an inlet cross-section of $2.71\,\mathrm{m} \times 1.81\,\mathrm{m}$ (width $\times$ height). Compared to a full scale wind turbine, the model size is scaled down at a geometrical scaling ratio of approximately 1:100 resulting in a mismatch in Reynolds number in the model experiment. The turbine is operated at a Reynolds number of approximately $Re_{\mathrm{tip}} \approx 10^5$ at the blade tip, which is more than one full order of magnitude lower than for full scale turbines. $Re_{\mathrm{tip}}$ is based on the chord length at the blade tip and the effective velocity during turbine operation.

Furthermore, the rotor swept area of the turbine model blocks 12.8 % of the wind tunnel's cross sectional area. The wind tunnel height is approximately twice the rotor diameter while its width measures about three times the diameter. Consequently, there is about one full diameter of space for lateral wake deflection on each side behind the rotor. However, an influence of the wind tunnel walls on the wake expansion and deflection cannot be completely excluded.

**Inflow conditions**

The measurements are performed for three different stationary inflow conditions as listed in Table 1. As shown in Figure 1 inflows B and C are generated by static grids at the inlet. The streamwise mean velocities and turbulence intensity levels measured in the empty wind tunnel at the turbine position ($x/D$=0) and wake measurement locations ($x/D$=3 and $x/D$=6) are presented in Figure 2.




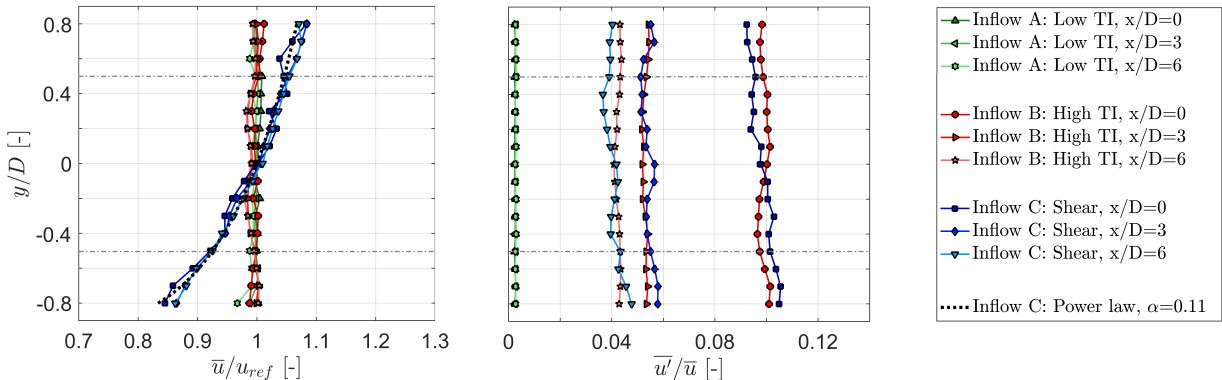

**Figure 2.** Normalized mean velocity $\overline{u}/u_{ref}$ and turbulence intensity $\overline{u'}/\overline{u}$ measured in the empty wind tunnel at the turbine position $x/D = 0$ and wake measurement positions $x/D = 3$ and $x/D = 6$

Inflow A can be characterized as a typical laboratory flow, in which the turbine is exposed to the uniform, low turbulence inflow of the wind tunnel ($TI_A$=0.23%). The low turbulence level in test case A is considered to be far below the intensities present in the real atmospheric boundary layer. Nevertheless, test case A is considered an extreme test case for the performance of computational prediction models. In order to generate a higher turbulence level for inflow B, a custom-made turbulence
grid with evenly spaced horizontal and vertical bars is placed at the test section inlet $x/D$=-2 upstream of the turbine. At the turbine position ($x/D$=0) a mean streamwise turbulence level of $TI_B$=10.0% is measured, which decays to 5.5% at $x/D$=3. Test case B represents turbulence conditions that are comparable to those of a neutral atmospheric boundary layer, although the inevitable decay of the grid-generated turbulence in the experiment is not representative for real conditions. Over the rotor swept area, inflow A is measured to be uniform within ±0.8% in $y$- and $z$-direction for all downstream distances. For inflow
B, wakes of the single grid bars are still observed at $x/D$=0, causing a spatial mean velocity variation within ±2.5%, while already at $x/D$=3 the grid-generated turbulent flow is uniform within ±1.0%.

The non-uniform shear inflow C is created by a grid with non-uniformly spaced horizontal bars, which is described in more detail in Bartl and Sætran (2017). The vertical flow profile establishes for all streamwise positions and can be approximated by the power law

$$\frac{u}{u_{ref}} = \left(\frac{y}{y_{ref}}\right)^{\alpha} \tag{1}$$

in which $\alpha$ describes the strength of the shear profiles gradient $du/dy$. The grid generated shear flow is approximated by a shear coefficient of $\alpha = 0.11$. Combined with a turbulence intensity of $TI_C$=10.0%, inflow C resembles conditions measured at an onshore site for a neutral atmospheric boundary layer (Wharton and Lundquist, 2012). In the $z$-direction, inflow C is measured to be spatially uniform within ±1.0% over the rotor-swept area. The v-component of the flow is observed to be slightly negative
for inflow C ranging from $v/u_{ref}$=[-0.005 -0.080] for all measurement positions. The influence of the negative v-component in the inflow is deemed insignificant for the streamwise velocity $u/u_{ref}$ in the wake. For the analysis of three-dimensional flow



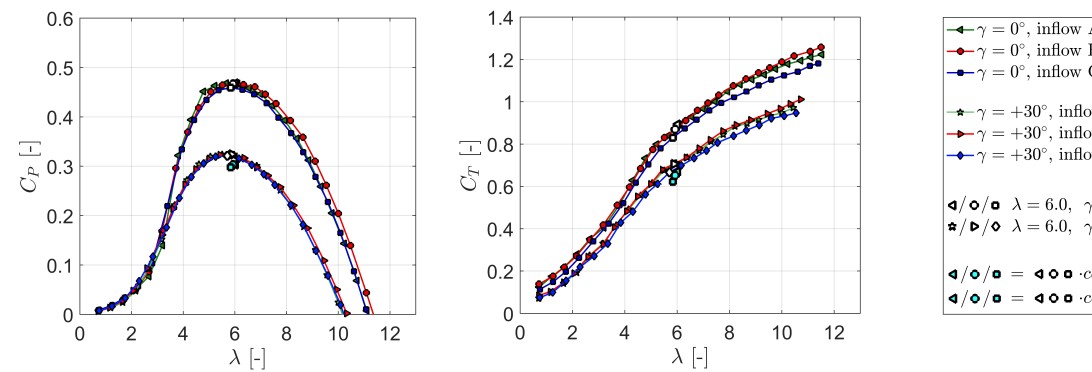

**Figure 3.** Operating conditions of the model wind turbine: **(a)** power coefficient $C_P$ and **(b)** thrust coefficient $C_T$ for different turbine yaw angels and inflow conditions. The white points indicate the operational conditions, at which wake measurements are performed. Cyan colored points indicate a theoretical power and thrust reduction by yawing of $C_{P,\gamma=0} \cdot cos^3(30°)$ respectively $C_{T,\gamma=0} \cdot cos^2(30°)$.

**Table 2.** Turbine performance ($C_P$ and $CT$) at the optimal operating point ($\lambda = 6.0$) for different yaw angles and inflow conditions

| | Inflow A | | Inflow B | | Inflow C | |
|---|---|---|---|---|---|---|
| $\gamma$ [°] | $C_P$ [−] | $C_T$ [−] | $C_P$ [−] | $C_T$ [−] | $C_P$ [−] | $C_T$ [−] |
| 0 | 0.468 | 0.893 | 0.467 | 0.870 | 0.459 | 0.830 |
| +30 | 0.322 | 0.707 | 0.324 | 0.706 | 0.321 | 0.667 |
| -30 | 0.328 | 0.711 | 0.331 | 0.713 | 0.327 | 0.679 |

effects in the wake the v-component from the inflow is subtracted. All presented mean velocity profiles and turbulence levels are measured in the empty wind tunnel at the reference velocity of $u_{ref} = 10.0m/s$.

**Operating conditions**

5   Figure 3 shows the turbine's measured power and thrust curves for different inflow conditions and yaw angles γ=0° and γ=+30°. In general, power and thrust measurements show very similar behavior for all three inflow conditions as shown in Table 2. Minor differences in the performance curves occur in the transition from stall around λ=3 as previously discussed in Bartl and Sætran (2017).

Performance curves measured for γ=-30° match well with those of γ=+30°, but are not plotted for clarity. For this study,
10   the turbine tip speed ratio is kept constant at its design point at $\lambda_{opt} = 6.0$ for all yaw angles and inflow conditions. For the investigated yaw angles $\gamma = \pm30°$ the power reduces about 30% compared to the maximum power of the non-yawed turbine. An approximation of this reduction can be obtained with sufficient accuracy by multiplying the maximum power of the non-yawed turbine by $cos^3(30°)$. An adequate estimate of the thrust coefficient of the yawed rotor can be obtained assuming a reduction by $cos^2(30°)$ on the thrust of the non-yawed rotor. This corresponds well to previous measurements by Krogstad and
15   Adaramola (2012).

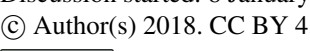





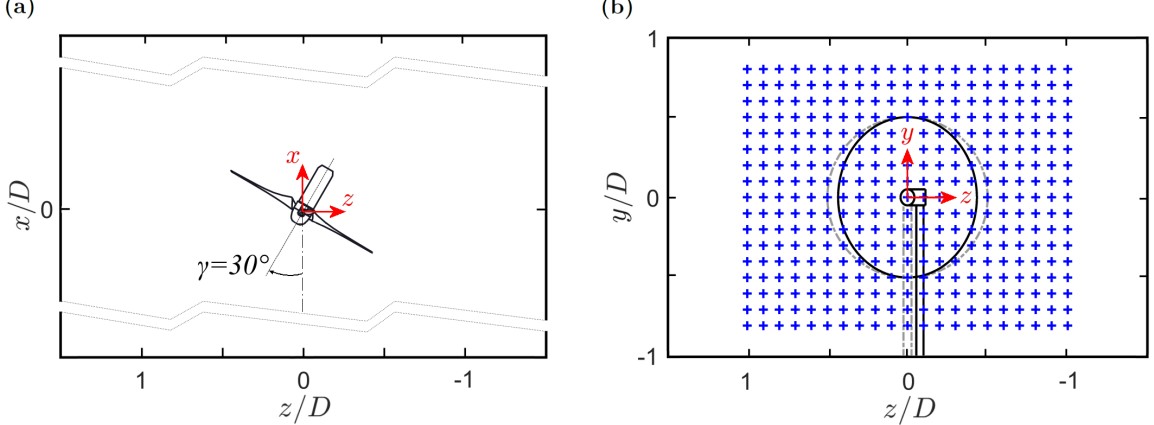

**Figure 4.** Reference coordinate system in the wind tunnel: **(a)** top view of yawed turbine setup and **(b)** grid for wake measurements.

## 2.2 Measurement techniques

### Power and force measurements

In order to assess the rotor power characteristics, the rotor was installed at another test rig equipped with a HBM torque transducer of the type T20W-N/2-Nm. Flexible couplings connect the torque transducer to the rotor shaft. An optical photo cell
5  is installed on the shaft enabling to measure the rotor rotational speed. The friction in the ball bearing between the rotor and torque sensor is measured without the rotor and thereafter subtracted from the total mechanical power. For the wake measurements the rotor is then installed on a smaller nacelle, which interacts less with the flow. The rotational speed is controlled via a servo motor, ensuring the same power characteristics. For measurements of rotor thrust the model turbine is installed on a six-component force balance produced by Carl Schenck AG.

### Flow measurements

The wake flow was measured with a two-component DANTEC FiberFlow Laser Doppler Anemometer (LDA) system used in Differential Doppler Mode. The laser was set up to record the streamwise flow component $u$ as well as the vertical flow component $v$. In order to obtain results for the lateral flow component $w$, the laser was turned in $u - w$ direction for one wake
15  measurement. The reference coordinate system and measurement grid is shown in Figure 4. $5 \times 10^4$ samples are taken for each measurement point over a period of approximately $30s$, resulting in an average sampling frequency of $1666Hz$. A grid consisting of 357 points is scanned for one full wake contour. For that purpose the LDA system is traversed from -1.0$D$ to +1.0$D$ in z-direction and from -0.8$D$ to +0.8$D$ in $y$-direction. The distance between two measurement points is 0.1$D$. For further analysis, these values are interpolated to a finer grid of $401 \times 321 \approx 129000$ grid points. The natural neighbor interpolation
20  method is used, which gives a smoother interpolation of the value distribution according to Sukumar (1997).





## 2.3 Measurement uncertainties

The uncertainty of the measured mean velocity is assessed for every sample following the procedure described in Wheeler and Ganji (2004). The LDA manufacturer Dantec Dynamics specifies the uncertainty on measured velocity by 0.04%. Random errors are computed from repeated measurements of various representative measurement points based on a 95 % confidence

interval. In the freestream flow as well as in the wake center the calculated uncertainties are below 1%, while increased uncertainties of up to 4% are calculated in the shear layers. Small inaccuracies in the adjustment of the traversing system are deemed to be the main contributor. The uncertainty in turbulent kinetic energy is computed according to the method proposed by Benedict and Gould (1996). Corresponding to the mean velocity the highest uncertainties up to 5% are found in the shear layer between wake and free stream flow.

## 3 Methods

### 3.1 Wake shape parametrization

In order to compare the shape of the mean wake for different inflows, the velocity contours are parametrized. The wake contours are therefore sliced into horizontal profiles for each of the 321 interpolated vertical positions. 201 of these 321 velocity profiles are located behind the rotor swept area from $y/D$=-0.5 to $y/D$=0.5. These profiles are fitted with a eighth order polynomial

to smoothen out local unsteadinesses. Then, an algorithm is applied to locate the $z$-position of the minimum fitted velocity for each profile. When plotting the $z$-positions of these all these minima versus their $y$-position, an arc shaped curve is obtained as a result. The curves allow for a direct wake shape comparison depending on inflow condition and yaw angle.

### 3.2 Wake deflection assessment

As intentional yaw misalignment could possibly be utilized for optimized wind farm control, an exact quantification of the

inflow-dependent wake deflection is an important input parameter. However, several methods to quantify the wake deflection have been used in the past, showing a large method-dependent variation in the deflection. Some of these methods are discussed in Section 5. In the present study an available power approach is used, which is deemed to give an solid assessment of the wake deflection. In order to assess the deflection of the wake, the potential power of an imaginary downstream turbine for various lateral offset positions is calculated. The $z$-position, at which the available power $P^*$ is minimum, is then defined as

the position of wake center deflection $\delta(z/D)$. In this study the available power $P^*$ is calculated for 50 different locations ranging from $-0.5 \leq z/D \leq 0.5$. The details of the method including an illustration are described in Schottler et al. (2017b).





**Figure 5.** Normalized mean velocity components $\overline{u}/u_{ref}$ for all measured yaw angles $\gamma = [-30, 0, +30]^\circ$, downstream distances $x/D$=[3, 6] and inflow conditions [A, B, C].



# 4 Results

## 4.1 Mean wake flow

At first the mean wake flows for all three yaw angles $\gamma = [-30, 0, +30]°$, both downstream distances $x/D$=[3, 6] and all three inflow conditions [A, B, C] are analyzed. Full cross-sectional wake measurements are presented in Figure 5. At the top the

wake flow for inflow A ($TI_A$=0.23%) is presented. The velocity deficit in the wake is observed to reduce significantly when the turbine is yawed. As the rotor thrust is reduced a smaller amount of streamwise momentum is lost in $x$-direction. For a yawed rotor a cross-stream momentum in $z$-component is induced. Due to this lateral force component the wake flow is deflected sideways. This is clearly observed at $x/D$=3, where the wake is seen to be deflected. Comparing the wake contours at $\gamma$=-30° and $\gamma$=+30° an asymmetry in the mean velocity distribution is obvious. The asymmetry between positive and negative wake

deflection is even more pronounced at $x/D$=6, where the wakes are seen to form a kidney shape. Both wake deflection and location of maximum velocity deficit are not symmetric, which is analyzed in more detail in the following sections.

### Effects of inflow turbulence

In the center of Figure 5 the mean velocity results of test case B, in which the inflow turbulence level is increased to

$TI_B$=10.0%, are shown. Due to a faster wake recovery the velocity deficits are observed to be smaller for all yaw angles. Increased turbulent mixing smoothened out the gradients between wake and freestream flow compared to test case A. The general wake shape and its lateral deflection for $\gamma = \pm 30°$ is seen to be similar as for the low turbulence inflow. A curled kidney-shaped velocity deficit is also observed at $x/D$=6 for test case B; however, the curl is not as pronounced as in test case A. Increased mixing might have smoothened the strong gradients in cross-flow direction in this case. The wake behind a

positively and negatively yawed turbine appears to feature a higher degree of symmetry than in test case A. Yet an asymmetry of the minimum wake velocity is still obvious for the increased background turbulence level in test case B.

### Effects of inflow shear

The wake results for a turbine exposed to inflow shear are shown at the bottom of Figure 5. The turbulence level $TI_C$=10.0%

is the same as in test case B, but shear is present in the inflow. Despite the sheared inflow the wake shapes for all three yaw angles and both downstream distances are observed to be very similar to those of test case B. The normalized velocity levels as well as the inner structure of the wake are almost identical. In the freestream region outside the wake the shear is clearly visible, especially the lower half. Compared to test case B, the wake of the tower is detectable in test case C. The tower wake recovery seems to be slower as the freestream fluid near the tunnel floor contains less kinetic energy in test case C.


### Curled wake shape

At $x/D$=6 a kidney-shaped velocity deficit is observed, showing a higher local velocities behind the rotor center. In other words, the maximum wake deflection is found at hub height. The curled kidney shape of the wake can be explained by the formation of a counter-rotating vortex pair, which was previously discussed by Howland et al. (2016) as well as Bastankhah and Porté-Agel





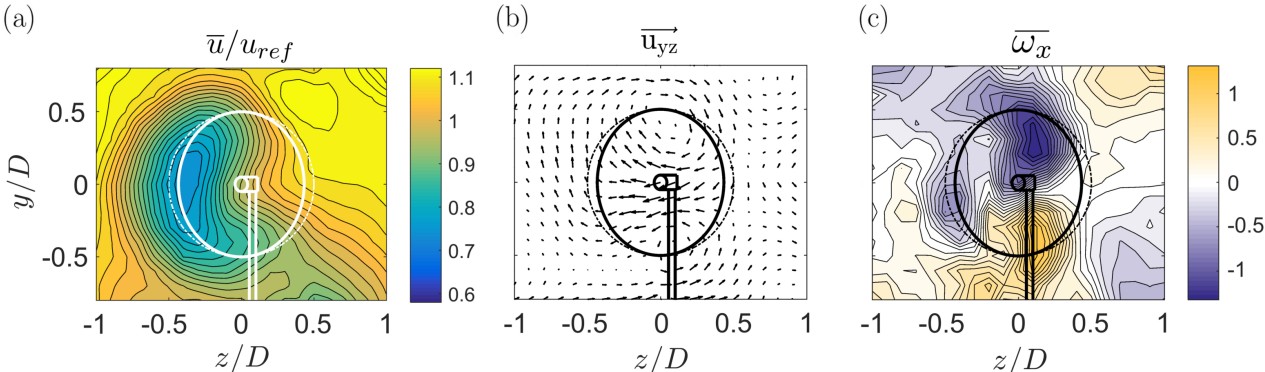

**Figure 6. (a)** Streamwise mean velocity $\overline{u}/u_{ref}$, **(b)** velocity vector $\boldsymbol{u_{yz}}$ in the yz-plane and **(c)** streamwise mean vorticity $\overline{\omega_x}$ at $\gamma = 30°$ and $x/D = 6$ at inflow C.

(2016). Bastankhah and Porté-Agel also presented a comprehensive explanation by the means of the differential form of the continuity equation. An illustration of the counter-rotating vortex pair at $x/D$=6 is presented in Figure 6, where the velocity vector $\overrightarrow{u_{yz}}$ as well as the mean streamwise vorticity $\overline{\omega_x}$ are calculated from all three velocity components. The velocity vector in the yz-plane is defined as $\overrightarrow{u_{yz}} = (v, w)$, while the streamwise time-averaged vorticity is defined as $\overline{\omega_x} = \partial v/\partial z - \partial w/\partial y$.

As shown in terms of $\overrightarrow{u_{yz}}$ the two vortex centers are formed approximately at the lower and upper boundary of the rotor swept area. The clockwise rotating vortex meets the counter-clockwise rotating vortex in the center behind the wake, leading to strong lateral velocities deflecting the wake sideways.

The locations of high rotation are furthermore visualized by increased levels of vorticity $\overline{\omega_x}$ around the vortex centers. The phenomenon of a counter-rotating vortex pair is not limited to rotating wind turbines. Howland et al. (2016) detected the simi-

lar large-scale vortices behind a non-rotating drag disc. Counter-rotating vortex pairs have previously investigated for jet flows exposed to a cross-flow e.g. by Cortelezzi and Karagozian (2001), a phenomenon which can be interpreted is the inverse to the wake flow behind a skewed rotor. In both phenomena the free shear flow, i.e. a wake or a jet, is superimposed with a strong lateral cross-flow, leading to the formation of a counter-rotating vortex pair at higher downstream distances.

**Tower wake deflection**

On the bottom of the wake contour plots, the wake of the turbine tower is indicated. The tower wake is observed to be deflected in the opposite direction than the rotor wake when the turbine is yawed. The deflection of the tower wake in the opposite direction is believed to have two reasons. Firstly, the turbine tower has a slight offset from $z/D$=0 as the center of yaw-rotation was set to the rotor midpoint and not the tower. Secondly, the tower wake experiences an additional deflection in opposite direction

due to an adversely directed cross-flow component outside near the wind tunnel floor (Figure 6 (b)). This cross-flow balances the counter-rotating vortex pair above and possibly deflects the tower wake further to the side.

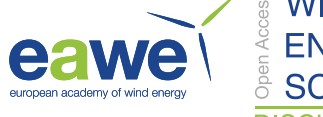

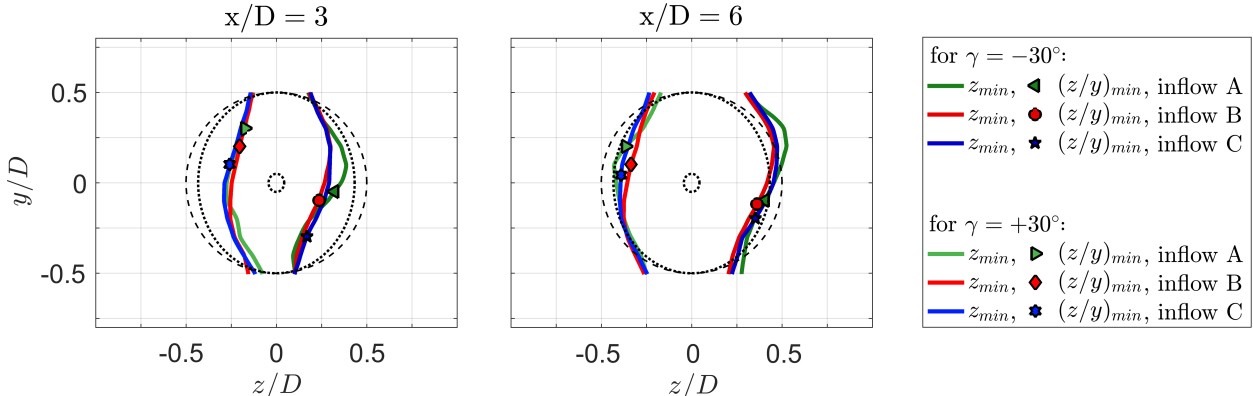

**Figure 7.** Minimum values in streamwise velocity $\overline{u}/u_{ref}$. Curl shapes and minimum positions are presented at $x/D$=3 (left) and $x/D$=6 (right) for the three different inflow conditions.

**Wake symmetry**

For the purpose of wake curl parametrization and wake symmetry assessment, the minimum values in streamwise velocity $\overline{u}/u_{ref}$ are extracted from the fitted wake contours for each vertical position ranging from $y/D$=[-0.5, ..., 0.5]. The detailed method is described in Section 3.1. This results in the $z_{min}$ lines as presented in Figure 7, which indicate the inflow-dependent wake curl. In addition to that, the position of the minimum velocity $(z/y)_{min}$ in both $y$- and $z$-direction is extracted and depicted in the plot by different symbols. The $z_{min}$ lines for all inflow conditions are observed to be slightly tilted in clockwise direction for both downstream distances $x/D$=3 and $x/D$=6. The counter-clockwise rotating turbine induces an initial clockwise rotation to the wake flow. Superimposing the clockwise wake rotation with the counter-rotating vortex pair thus results in a slightly tilted curled wake shape. As previously mentioned the wake curl is seen to be more asymmetric for the low background turbulence test case A. A significant bulge is visible for $\gamma$=-30° in the upper half of the wake for both downstream positions. For inflow conditions B and C the curl parametrization lines are almost coinciding, confirming the insignificant influence of the moderately sheared inflow on the wake shape. Analyzing the locations of minimum velocities $(z/y)_{min}$ in the wake contours, a deviation from the horizontal centerline $y/D$=0 in for both positive and negative yaw angles is obvious. For $\gamma$=-30° the minimum velocities $(z/y)_{min}$ are deflected to the lower half of the wake, while an upward deflection happens for positive yaw angles $\gamma$=+30°. In agreement with Bastankhah and Porté-Agel (2016), the wake rotation is assumed to turn the velocity minimum in clockwise direction initially. The deflection from the wake centerline is observed to be larger for $x/D$=3 than for $x/D$=6, where mixing processes already have smoothened the gradients. In the case of sheared inflow of test case C, the locations of minimum wake velocity $(z/y)_{min}$ are found to be lower than for test cases A and B.

**Wake center deflection**

The 3D Available power method introduced in Section 3.2 and a 2D Gaussian fit method are used for the quantification of wake deflection. In order to judge possible blockage effects, another rotor of a smaller diameter ($D_{Rot,small}$=0.45 m,



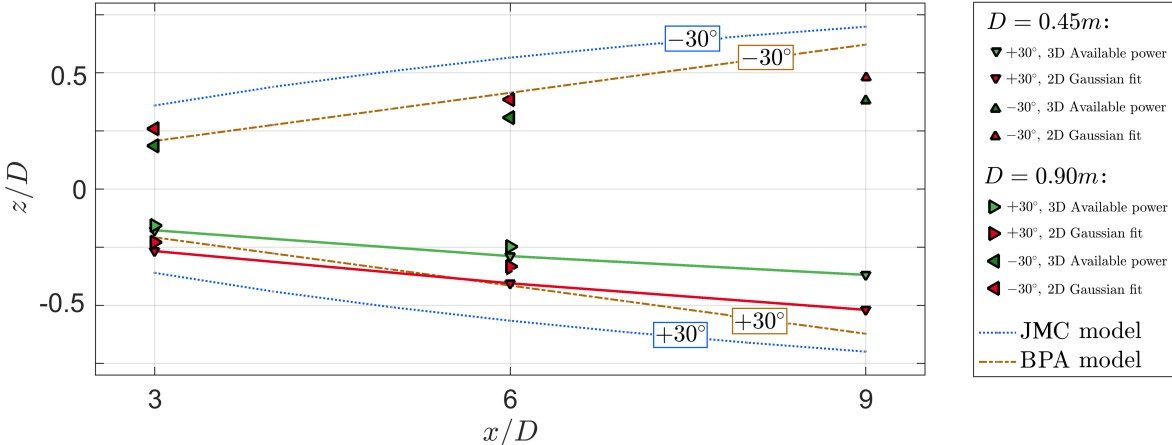

**Figure 8.** Calculated wake deflection $\delta(z/D)$ for the NTNU rotor ($D = 0.90m$), a downscaled NTNU rotor ($D = 0.45m$) as well as Jiménez et al.'s and Bastankhah and Porte-Agél's wake deflection model. The inflow turbulence level is $TI_A = 0.23\%$.

$\sigma_{Blockage,small} = \frac{A_{Rot,small}}{A_{Tunnel}}$=3.3%) was used in addition to the 0.90 m ($\sigma_{Blockage,large}$=12.8%) rotor. The details of the experimental setup featuring the smaller 0.45 m rotor are described in Bartl et al. (2018). Further, the results are compared with two different wake models by Jiménez et al. (2010) (JCM) and Bastankhah and Porté-Agel (2016) (BPA). The comparison of the wake deflections are shown in Figure 8. At $x/D$=3 the wake deflection for $\gamma$=+30° of the smaller rotor and the original

rotor match very well. At $x/D$=6 a small deviation in the wake deflection after the rotors of different sizes and blockage is calculated. It can be assumed that blockage by the wind tunnel walls influences the wake deflection; however, the difference in deflection between the different rotors is observed to be rather small. Comparing the measured deflection with the prediction models discloses larger deviations. The deflection predicted by the JCM-model is generally observed to be larger than the one predicted by BPA-model. The calculated wake deflection by the Available power method at $x/D$=3 is still well predicted by

the BPA-model, while more significant deviations are observed at $x/D$=6. Obviously larger differences in wake deflection are predicted by the JCM-model, both at $x/D$=3 and $x/D$=6. A number of reasons are possible to cause the significant deviations between measured and modeled deflection results. Besides the discussed wind tunnel blockage, a major source of uncertainty in this comparison arises from the method used to calculate the wake deflection. Quantifying the wake deflection by he minimum of a fitted Gaussian on the hub height velocity profiles results in a better match with the BPA-model at $x/D$=6 as shown

by the red curve in Figure 8. However, using the hub height profile only for the wake center deflection does not take the total mean kinetic energy content in the wake into account. Due to the complex three dimensional shape of the velocity deficit, a reduction of the wake deflection to one single value has been shown to be complex. A number of different methods have been proposed, resulting in many different deflection quantifications (Vollmer et al., 2016). Secondly, the wake deflection $\delta(z/D)$ for all three inflow conditions is compared. These results are shown in Figure 9 and compared to the BPA-model. In contrast

to the JCM-model, the inflow turbulence intensity is a input variable in the BPA-model. It can be observed that the BPA-model predicts a higher wake deflection for a smaller inflow turbulence level. Bastankhah and Porté-Agel (2016) argue that smaller

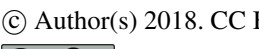



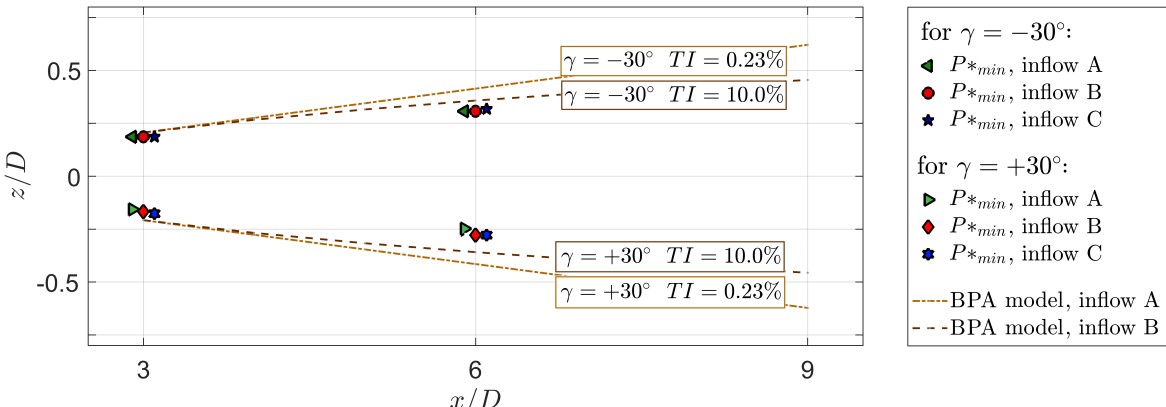

**Figure 9.** Calculated wake deflection $\delta(z/D)$ at $x/D$=3 and $x/D$=6 for three different inflow conditions A, B and C compared to TI-dependent deflection predictions by Bastankhah and Porte-Agél's wake deflection model. Note that a small offset in $x/D$ of the measured values was chosen for better visibility.

**Table 3.** Lateral deflection $\delta(z/D)$ [$-$] and wake skew angle $\xi$ [$°$] calculated with the available power method.

| | | Inflow A | | Inflow B | | Inflow C | |
|---|---|---|---|---|---|---|---|
| $\gamma$ [$°$] | $x/D$ [$-$] | $\delta(z/D)$ | $\xi$ [$°$] | $\delta(z/D)$ | $\xi$ [$°$] | $\delta(z/D)$ | $\xi$ [$°$] |
| 0 | 3 | 0.015 | 0.29 | 0.005 | 0.10 | 0.015 | 0.29 |
| +30 | 3 | -0.157 | -2.99 | -0.167 | -3.18 | -0.177 | -3.38 |
| -30 | 3 | 0.187 | 3.57 | 0.187 | 3.57 | 0.187 | 3.57 |
| 0 | 6 | 0.026 | 0.24 | 0.036 | 0.34 | 0.036 | 0.34 |
| +30 | 6 | -0.248 | -2.36 | -0.278 | -2.65 | -0.278 | -2.65 |
| -30 | 6 | 0.308 | 2.94 | 0.308 | 2.94 | 0.318 | 3.03 |

inflow turbulence reduces the flow entrainment in the far wake and thus increases the wake deflection. The calculated lateral deflection values $\delta(z/D)$ and the associated wake skew angle $\xi$ are furthermore listed in Table 3.

In general, a very similar wake deflection is observed for all three inflow conditions at both downstream distances. A
5 systematic asymmetry in the wake deflection behind a turbine yawed $\gamma$=-30° and $\gamma$=+30° is observed. The wake shows a higher deflection for negative yaw angles in all inflow cases. Also the wake behind the non-yawed turbine is seen to be slightly deflected in positive z-direction, which is assumed to stem from the interaction of the rotating wake with the turbine tower. As discussed by Pierella and Sætran (2017) who performed experiments on the same rotor with a larger tower, the tower-wake-interaction can lead to an uneven momentum entrainment in the wake. Increasing the turbulence level from $TI_A$=0.23% to
10 $TI_B$=10.00% is found to only have a small influence on the wake deflection. In fact, no difference is detected for $\gamma$=-30°. For





$\gamma$=+30°, however, a slightly smaller wake deflection is calculated for the lower inflow turbulence. This can also be interpreted as a higher degree of asymmetry for low background turbulence. Adding shear to the inflow is not observed to change the wake deflection significantly. This confirms the above-mentioned similarity in wake shapes measured for test cases B and C.

## 4.2 Rotor-generated turbulence

For the measurements presented in the this study the kinetic energy is considered to be fully dominated by turbulent motions from $x/D \geq 3$ for inflow A, as Eriksen and Krogstad (2017) recently showed that the production of rotor-generated turbulent kinetic energy is finished at $x/D$=3 for measurements on the same rotor and inflow condition. For inflow conditions B and C, the transition to fully turbulent motions is expected to take place at even smaller downstream distances.

**Effects of yawing on turbulent kinetic energy locations**

At the top of Figure 10 the TKE levels in the wake are presented for test case A ($TI_A$=0.23%). As observed in earlier studies (Bartl and Sætran, 2017; Eriksen and Krogstad, 2017) a ring of high turbulence levels is formed behind the tips of the rotor blades for a non-yawed turbine. In this region the tip vortices decayed and into turbulent motions. With increasing downstream distance the sharp peaks decrease in magnitude and blur out to their surrounding. For a yawed turbine, the ring of peak turbulence is laterally deflected and deformed accordingly. For $x/D$=3 the peaks are clearly separated by an area of low turbulence in the center of the deflected wake. For $x/D$=6, this area is observed to be significantly smaller. The peaks are still distinct, but it is expected that they start merging into one peak for higher downstream distances. The strongest TKE levels are observed for locations of the highest gradient in mean streamwise velocity. Thus, the TKE-ring's extension is observed to be slightly larger than the contours of the mean streamwise velocity, emphasizing the need to take the parameter TKE into account in wind farm site planning or yaw control studies.

**Effects of inflow turbulence and shear**

The TKE contours for increased inflow turbulence of test case B are shown in the center of Figure 10 as well as the red lines in Figure 11. At $x/D$=3, slightly smaller TKE peaks and higher centerline turbulence are measured for test case B than for test case A. The higher TKE levels in the freestream lead to an increased mixing, which is reducing the TKE peaks in the tip region. At $x/D$=6 the TKE peaks are observed to be at about the same level for both inflow conditions. For the yawed cases also the turbulence level in the wake center has evened out between inflow cases A and B. The TKE levels for the sheared inflow in test case C are observed to be very similar to those of test case B for all investigated yaw angles. These findings suggest that the presence of a moderate shear flow in a highly turbulent boundary layer does not influence the production of rotor-generated turbulent kinetic energy significantly.

**Effects of yawing on turbulent kinetic energy levels**

The levels of peak turbulence are observed to decrease considerably when the rotor is yawed. For a direct case-to-case comparison, TKE-profiles at hub height $y$=0 at $x/D$=6 are presented for $\gamma = 0°$ and $\gamma = -30°$ in Figure 11.





**Figure 10.** Turbulent kinetic energy $k/u_{ref}^2$ for all measured yaw angles $\gamma = [-30, 0, +30]°$, downstream distances $x/D=[3, 6]$ and inflow conditions [A, B, C]





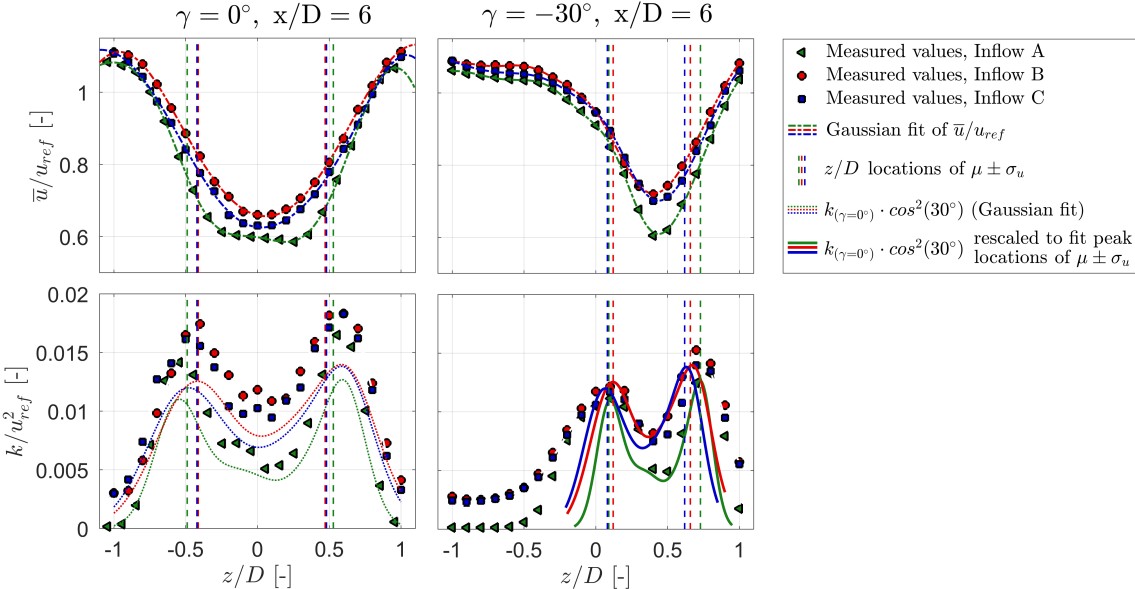

**Figure 11.** Normalized mean velocity and turbulent kinetic energy $k/u_{ref}^2$ profiles at hub height $y = 0$ and $x/D$=6. The yaw angles are set to $\gamma = 0°$ and $\gamma = -30°$. Vertical dashed lines give the borders of standard deviations of fitted velocity profiles $\mu \pm \sigma_u$. Dotted lines indicate a TKE profiles at $\gamma = 0°$ multiplied by $cos^2(\gamma = -30°)$. Full lines have the same magnitude as the dotted lines, but are linearly scaled in $z$ to fit the peak locations of $\mu \pm \sigma_u$.

For a yawed turbine, the rotor thrust reduces with approximately $cos^2(\gamma)$ as previously shown in Figure 3. Multiplying also the TKE levels generated by the non-yawed rotor with $cos^2(\gamma)$ is observed to result in a decent first order approximation of the turbulence levels behind the yawed rotor. The reduced TKE levels for $\gamma = -30°$ are indicated by the dotted lines in Figure 11. In order to also find the lateral deflection of the turbulence peaks for yawed rotors, another first order approximation of their

5  location proposed by Schottler et al. (2017a) is applied. In this approach the expected value and standard deviation of the fitted velocity profile behind a yawed rotor is calculated. Adding the standard deviation to the expected value $\mu \pm \sigma_u$ gives a rough estimate of the corresponding TKE peak locations, as shown by the vertical dashed lines in Figure 11. Thus, it is possible to rescale the TKE peak locations and levels by knowing TKE and mean velocity for the now-yawed case. This might be a useful addition for modeling the rotor-generated turbulence in yawed wakes. For a complete assessment of mean velocity and turbu-

10  lent kinetic energy in a yawed wind turbine wake, the model for streamwise velocity profiles by Bastankhah and Porté-Agel (2016) could be extended by the proposed relations for the rotor generated turbulence.





## 5   Discussion

The present wind tunnel investigation showed detailed flow measurements in the wake of a yawed model turbine for different inflow conditions. A number of modeling techniques and turbine sizes were used in previous yaw wake studies in the literature, resulting in a significant variation in wake shapes and their deflection. However, a number of general flow effects in the wake
behind a yawed turbine seem to be reproducible

Our results indicated minor asymmetries in the wake flow behind positively and negatively yawed turbines. The interference of the modified flow field around the tower and wake rotation is deemed to be the source for this asymmetry. This explanation is consistent with findings by Grant and Parkin (2000), who reported clear asymmetries in the tip vortex shedding and circulation in the wake for positive and negative yaw angles. Our experimental measurements showed a kidney shaped mean
velocity deficit at $x/D$=6 for all inflow conditions. These results agree well with recently discussed experimental results by Howland et al. (2016). Although the wake shape was not specifically discussed, a curled wake shape was already indicated in the results presented by Medici and Alfredsson (2006). The results presented by Bastankhah and Porté-Agel (2016) offer a good comparison as wakes were measured at a number of yaw angles and downstream distances. The wake shape and velocity deficit at $\gamma = \pm 30°$ and $x/D$=6 match qualitatively well with our results, when an opposite sense of turbine rotation is taken
into account. A direct comparison of the wakes at $x/D$=3 and $x/D$=6 of the here presented results of test case B with an equivalent setup for a slight smaller model turbine of different rotor geometry was performed by Schottler et al. (2017c) and Schottler et al. (2017a). These results show a more distinct curl in the wake already at $x/D$=3 while velocity deficit and wake deflection are generally found to be very similar for both model turbines.

Our study moreover indicates that the wake shape and deflection is affected by inflow turbulence. This confirms the imple-
mentation of the inflow turbulence as an input parameter in the recently developed wake model by Bastankhah and Porté-Agel (2016). The influence of the inflow turbulence seems to be slightly overpredicted by their model, although a more thorough analysis for different yaw angles and downstream distances on a smaller, unblocked rotor are needed for a solid assessment of the model's sensitivity to inflow turbulence. Furthermore, the comparison of the model-predicted deflection and experimentally obtained results is not straightforward. Due to the various different calculation methods used the assessment of the wake center
deflection is found to be equivocal. Gaussian fitting to locate the minimum wake velocity was amongst others used by Jiménez et al. (2010) as well as Fleming et al. (2014), while Luo et al. (2015) and Howland et al. (2016) calculated the center of mass of the three-dimensional velocity contour. A comparison of different wake deflection methods was presented by Vollmer et al. (2016), showing the significant method-related variation in deflection quantification.

Another focus of the present study was to assess whether the wake's properties are significantly influenced by sheared inflow.
Shear is present in most atmospheric boundary layer flows and highly dependent on stability and the terrain's complexity and roughness. The strength of the investigated shear in test case C is rather moderate and considered typical for a neutral atmospheric boundary layer (Wharton and Lundquist, 2012). As the study investigated only two different shear flows ($\alpha_B$=0.0 and $\alpha_C$=0.11), solid statements about the wake flow's sensitivity to this parameter cannot be made. The results do however indicate a rather insignificant effect of such a moderate shear on the wake flow. Possibly, a considerably stronger shear at lower inflow



turbulence would have resulted in more distinguishable wake characteristics. In contrast to a recent full-scale LES study by Vollmer et al. (2016), our results seem to shown a rather small dependency of the wake characteristics on the inflow conditions. However, Vollmer et al. (2016) varied four different inflow parameters (turbulence intensity, potential temperature wind shear and veer) simultaneously, which made direct conclusions on the sensitivity to a single inflow parameter difficult. In conclusion,

our results do not contradict with their findings as the inflow conditions in both setups were modeled very differently.

## 6   Conclusions

An experimental study on the inflow-dependent wake characteristics of a yawed model wind turbine was realized. In accordance with previous studies, it is confirmed that intentional turbine yaw misalignment is an effective method to laterally deflect

the velocity deficit in the wake and thus offers a large potential for power optimization in wind farms. For the equally important optimization of downstream turbine fatigue loads, a careful planning of wind farm layout and control strategy should thus also take the strength and expansion of rotor-generated turbulence footprints into account. It is shown that the rotor-generated turbulence distributions are deflected in the same degree as the mean velocity profiles, but feature a slightly wider expansion. Further analysis demonstrated that an increasing yaw angle reduces the levels of the peak turbulence, which is decreasing at a

similar rate as the rotor thrust.

The study moreover recommends a consideration of the inflow turbulence level as an important parameter for deflection models implemented in wind farm controllers, as it is affecting the yaw-angle dependent symmetry in shape and deflection. The degree of asymmetry was observed to be higher for lower inflow turbulence. The recently proposed wake deflection model by Bastankhah and Porté-Agel (2016) proved to deliver good approximations of inflow-turbulence-dependent wake deflection.

However, more wake measurements at different yaw angles and various downstream distances should be performed to fully assess the model's sensitivity to inflow turbulence. As the influence of a gentle inflow-shear on the wake characteristics was found to be insignificant, an inclusion of this parameter in wake models is thus not considered to be essential at this stage. The experimental results revealed very similar velocity deficit and rotor-generated turbulence distributions to those measured for an uniform inflow.


*Competing interests.*   The authors declare that there are no competing interests.

*Acknowledgements.*   The authors would like to acknowledge the IPID4all travel support granted by the German Academic Exchange Service (DAAD).




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
