# Peer review of "Wind tunnel experiments on wind turbine wakes in yaw: Effects of inflow turbulence and shear"

_Wind Energy Science, 2017_

## Referee Comment (RC1) · Anonymous Referee #1 · 30 Jan 2018

This paper provides a very useful study of wakes in yaw through a detailed and careful wind tunnel campaign. Presenting the results from the wind tunnel is valuable and confirm some assertions made by LES studies. The separation of inflows into low and high TI, and with shear, provides useful into the relationship between inflow and wake behavior. The comparisons of the wind tunnel experiments with well-known models is also a helpful analysis. Overall I found the paper to be well-written, the figures clear the arguments well structured and the contributions important.

Overall Comments:

Symmetry: Sometimes I became a little confused about discussions on symmetry. At some points (page 12 for example) the focus was on the shape of curl, but on bottom page 14, I had the impression symmetry here meant a difference in the effectiveness

of positive versus negative yaw. Maybe this could be further clarified.

Further, if I understand, both asymmetries are explained as being explained by inter-action with the tower. This made sense to me in the discussion of the symmetry of the wake itself, but I had some doubts if it could fully explain the asymmetry in +/- ef-fectiveness. For example, some LES codes show this asymmetry while not including any tower model in the flow (for example ALM, or ADM codes which have essentially only the rotor modeled). Wouldn't this imply some other mechanisms could also be responsible?

A final point on this discussion, could you include some discussion of the proximity of the rotor to the ceiling and the floor? I was thinking a source of discrepancy might be that LES/field data will have only a ground, and as a result only one of the vortices experiences ground effects. Is this a consideration?

A second overall comment, the authors point out that is difficult to reduce wake de-flection to a single value, and can complicate interpretation of results such as Fig 8-9. Since you already employ the method of available power, I believe an interesting ad-ditional comparison between the collected data and the models would be to compare the power output of an imaginary turbine located at x/D=6 and z/D=0 (and perhaps z/D = +/- 0.5). This could represent an interesting assessment of do the models correctly predict the change in power obtained through wake steering for a given arrangement.

Specific comments:

1) The introduction is well done, with a good review of the literature to date. Useful to read it summarized in this way.

2) The selection of y as vertical and z as cross-wise was surprising to me, although since you provide a coordinate system in Fig 4., not too confusing. But is there a reason for this? FAST and Bladed for example both have z directed upward

3) Page 6, cos cubed is found for power-loss function. Anecdotally, this would be high

for a utility-scale turbine I believe (although it fits the theoretical value). Is this a function of the scaling?

4) Fig 11: I didn't understand why for the lower plots, two different methods of fitting are used. It had the impact on me, to reemphasize the difference in value of the points, since on the right the higher points are outliers to the fit.

Connection to the companion paper:

I also could use a little more explanation of which material has been put into which paper and why. For example, the companion paper is focused on changes in TKE, and wind speed variability. Does it make sense to also discuss TI in this paper? To be clear, I am fine with the current division, but it would be helpful to understand a little more the distinction between the papers, if they both include profiles of turbulence for example. Perhaps one additional paragraph more explicitly delineating the papers, to be added to both?

---

## Referee Comment (RC2) · Anonymous Referee #2 · 26 Feb 2018

The paper covers the very relevant topic of wake steering, through a fundamental analysis of the wake physics behind the wind turbine in yaw. The detailed results of the cross-sections of the wake regarding velocity deficit and turbulent kinetic energy levels are a rich addition to our understanding of the complex wake aerodynamics involved in the wake steering process. The systematic investigation of flow cases with increased complexity (no turbulence, turbulence, turbulence with shear) is also very welcoming, as it allows to distinguish between the different phenomena which drive the wake deflection. The introduction is very well written and covers the state-of-the-art as it is today. An important remark is made about the triviality of wake deflection and wake expansion methods used in contemporary literature, and a well-considered attempt is made to base the definition of deflection on the integrated power levels in the wake.

[Figure]

This notion about definitions and parameters involved in wake deflection certainly requires a lot more discussion by the wind energy community. Below I have stated remarks related to the content of the report, as well as technical remarks including quite some spelling mistakes. Note, p99.l99 stands for page 99, line 99.

Remarks about the content:

In the introduction, p1.l20, you state wake redirection techniques, which intentionally apply an uneven load distribution. Instead of an uneven loading, I would say that key to wake steering is the tilting of the thrust vector. For instance, cyclic pitching results is a large uneven loading, but marginal steering, while yaw results in a much smaller uneven loading, but a large thrust vector tilting.

On page 11, in the subsection about the tower wake deflection, you discuss several factors that contribute to the tower shadow deflection. You mention the influence of the lateral offset between the rotor and the tower during yawing, and the effect of the CVP (Counter-rotating Vortex Pair) on the wake opposite tower wake direction. What I suspect here is that the bottom of the two counter-rotating vortices is in strong interaction with its mirror image underground (i.e. the ground effect), thereby forming another CVP, but in opposite direction to the main CVP involved in the wake steering. This could hypothetically boost the deflection of the wake shadow in opposite direction to the main wake deflection.

On page 12, the insignificant influence of the moderately sheared inflow on the wake shape is addressed. However, this can only be stated about the shear inflow under high turbulence conditions, as that is the only case you analysed. It might be the case that shear does have a significant contribution for low ambient turbulence levels, as the inflow shear in combination with the wake shear results in a distinctively high velocity gradient near the top of the wake (as shown by many researchers), thus increasing turbulence levels there. By the way, you mention this notion in the discussion about the TKE results later on in the paper.

For completeness, it is important that the parameter settings for the JMC and BPA models is provided.

On page 14, you note that the wake deflection of a non-yawed turbine is assumed to stem from the interaction of the rotating wake with the turbine tower. The fact that the wake of a counter-clockwise rotating turbine (thus with a clockwise rotating wake swirl) deflects in positive z direction, sounds to me as originating from the interaction between the wake swirl and the ground: the root vortex forms a CVP with its mirror image underground from the ground effect, with its deflection direction in positive z direction. This was also discussed by e.g. Fleming (2014) and BPA (2016).

On page 14, you mention that the differences are small for the wake deflection as compared between a high and low turbulence inflow. Here it would be helpful to present results of the streamwise vorticity for both cases and for several downstream positions. Maybe the diffusion of vorticity under self-induced turbulence is already very significant for low ambient turbulence levels, which would explain why both cases are then so similar. In the end, the analysis of streamwise vorticity is key to understand, as the streamwise vorticity forms the CVP which is the driving force behind both the wake deflection and the shape deformation.d

In figure 11, vertical lines of the standard deviation are given, but it is unclear how the mean and standard deviation are defined here. After all, the Gaussian fit curves applied here are clearly not symmetric, thus I assume those are from a fit with multiple Gaussians, for which it is less trivial to define a mean and standard deviation. Apart from that, there is a lot of information in this figure, and it took me a while to comprehend it fully.

Technical remarks:

P3.l4 – "donstream" -> "downstream" P3.l24 – remove "used in" P3.l27 – "a NREL" -> "an NREL" Table 1 – add full stop Figure 2 – add full stop Table 2 – "CT" -> "CÂňÂňT" Table 2 – add full stop P7.l3 – "a HBM" -> "an HBM" P8.l14 – "a eight" -> "an eight"

P8.l16 – remove "these" before "all these" P8.l16 – remove "as a result" (duplication with "is obtained") P9.l22 – "an solid" -> "a solid" P11.l10 – add "been" in between "previously investigated" P12.l13 – remove "in" P13.l9 – add "the" before "BPA-model" P13.l9 – lowercase for "Available" P13.l16 – duplicate of the word "complex" P13.l20 – "a input" -> "an input" P15.l13 – remove "and" Figure 10 – add full stop Figure 11 – remove "a" before "TKE profiles" P18.l16 – "slight" -> "slightly" P19.l2 – "shown" -> "show"

Comma's could be used more extensively to increase readability. For instance, see the first paragraph of section 4.1: "At the top, the"; "As the rotor thrust is reduced, a"; "For a yawed rotor, a"; "Due to this lateral force component, the"; "Comparing the wake contours [. . .] , an asymmetry". . .

Sometimes it would make the text more easily readable if the text would be broken up into several paragraphs. For instance, p13.l18: "Secondly, the wake. . ." is a confusing construction, as there is no "firstly" defined in your text. Moreover, this sentence refers to a new comparison, so to clarify the text it would be better to break it up into two sections.

At p2.l18, "The measured circulation in the wake showed clear asymmetries for positive and negative yaw angles". This is about the asymmetry of the wake regarding the kidney shape, but this sentence could also be read as an asymmetry between the values for positive and negative yaw (i.e. yaw dependency).

In figure 1, a clockwise rotating turbine is presented in the left subfigure, while the other two subfigures depict an anti-clockwise rotating turbine. Although it was clearly mentioned in the text that the results were for a turbine that is anti-clockwise rotating, it was a bit confusing for me at first to see the picture for the clockwise rotating turbine (which I assume is the second turbine used for the experiments later on in the paper).

At page 6, it would be good to add the approximate values for cos2(30) and cos3(30) in the main text to get a feeling for their magnitude (0.75 and 0.65 respectively).

In figure 5 you apply a very fine gradient scaling with contour lines added, but it is hard to extract the true magnitude from these plots. You might change these plots to one where you have much fewer gradient colors (let's say about 10), and change the colorbar accordingly (which is completely smooth in the current visualization).

In section 4.1, the subsection about the curled wake shape, you mention ". . . a kidney-shaped velocity deficit is observed. . .", without referring to a figure number. The same applies for the subsection about the tower wake deflection on the next page.

In general, it comes more natural for the understanding of the reader if the lateral direction was defined as y and the vertical direction as z instead.

---

## Author Comment (AC1) · 26 Mar 2018

**Authors' response to Referee #1:**

We would like to thank the referee for reviewing this manuscript, the valuable feedback and the very constructive comments. At this stage of the review process, we respond to the referee #1's comments and propose improvements for the final manuscript. The referee's original comments are printed in **bold** followed by the corresponding answers. Passages from the manuscript are printed in *italic writing*, in which proposed additions are indicated in blue and deleted parts in red. Theory were work for your efforts

Thank you very much for your efforts,

Jan Bartl on behalf of all authors

**Overall comment (1a)**

Symmetry: Sometimes I became a little confused about discussions on symmetry. At some points (page 12 for example) the focus was on the shape of curl, but on bottom page 14, I had the impression symmetry here meant a difference in the effectiveness of positive versus negative yaw. Maybe this could be further clarified.

Thank you for this very constructive comment. Indeed, the term 'symmetry' refers to two different parameters in those cases, which should be further clarified. On the top of page 12 the symmetry of the shape of wake curl is analyzed, while further down (bottom of page 12 to bottom of page 14) the symmetry in effective wake deflection is compared. In both cases, however, the symmetry is analyzed with respect to positive versus negative yaw angles. In the comparison on page 12, the three-dimensional wake scans behind a positively and negatively yawed turbine are parametrized to twodimensional curves showing local velocity minima. In the comparison on page 14, however, the three-dimensional wake scans are parametrized to a single value quantifying the overall wake deflection. For clarification, the following changes are suggested for the manuscript:

**p.12, l.1 ff:**

**Wake curl symmetry**

In order to compare the three-dimensional wake shapes behind a positively versus negatively yawed turbine more quantitatively, the curled shapes of the velocity deficit area are parametrized to a two-dimensional line. For this purpose, the minimum values in streamwise velocity  $\overline{u}/u_{ref}$  are extracted from the fitted wake contours for each vertical position ranging from y/D = [-0.5, ..., 0.5]. The detailed method is described in Section 3.1.

**p.12, l.20 ff:**

**$\overline{Overall w}$ ake $\frac{center}{center}$ deflection**

The 3D-three-dimensional Available power method introduced in Section 3.2 is used to quantify the overall deflection of the kinetic energy contained in the wake. As explained in Section 3.2 the minimum available power in a circular area in the wake is located, which is reducing the full wake flow field to a single parameter representing the overall wake deflection. A comparison of the minimum available power in the wakes behind a positively versus negatively yawed turbine enables a comparison of symmetry in the deflection of the energy contained in the wake with respect to the yaw angle. Additionally, a 2D-two-dimensional Gaussian fit method are used for the quantification of wake deflection. for the wake center detection at the turbine's hub-height is used to demonstrate systematic differences in the deflection quantification methods

**p.14, l.5 f:**

A systematic asymmetry in the wake deflection represented by the minimum available power behind a turbine yawed  $\gamma = -30^{\circ}$  and  $\gamma = +30^{\circ}$  is observed.

**Overall comment (1b)**

Further, if I understand, both asymmetries are explained as being explained by interaction with the tower. This made sense to me in the discussion of the symmetry of the wake itself, but I had some doubts if it could fully explain the asymmetry in +/- effectiveness. For example, some LES codes show this asymmetry while not including any tower model in the flow (for example ALM, or ADM codes which have essentially only the rotor modeled). Wouldn't this imply some other mechanisms could also be responsible?

Thank you for this very good comment. This is one of the very substantial questions that require to be clarified when discussing possible causes for deflection asymmetries during wake steering. Yes, we deem the interaction of the rotor wake and tower wake to be the main reason for the slight asymmetries in both the wake curl and also the resulting overall wake deflection. The tower structure and its wake introduce an asymmetry to the otherwise perfectly symmetrical setup. However, other mechanisms can potentially affect the wake deflection symmetry, especially in the case of full-scale turbines. These are discussed in the following:

Mechanisms that generally can introduce asymmetry to a yawed turbine setup:

- (1) non-uniform inflow to the rotor, e.g. shear or veer
- (2) ground effects/wall blockage effects
- (3) systematic errors in turbine yaw alignment
- (4) tower wake interaction

(1) The effects of a vertical sheared inflow on wake steering through yaw was recently investigated in an experiment by Schottler et al. (2017a). They found an asymmetric power distribution of an aligned downstream turbine with respect to the upstream turbine yaw angle, when a strong vertically sheared profile was present in the inflow. By inverting the vertical shear in the inflow, the power distribution of the downstream turbine was again asymmetric, however towards the opposite sign of the upstream turbine yaw angle.

Asymmetries in the deflection of the yawed wake are simulated in a LES by Vollmer et al. (2016), in which a combination of inflow shear and veer are deemed to be responsible for the asymmetric wake shapes especially in stable atmospheric conditions. An asymmetric combined power distribution is also observed in another LES study on full-scale turbines by Fleming et al. (2015), where the turbines are exposed to a LES-generated atmospheric boundary layer. Therein, Coriolis forces and wind veer are discussed as a reason for differences in wake deflection. In a recent follow-up study by Fleming et al. (2017) veer is kept to a minimum and no deflection of the non-yawed baseline case is observed. The deflection asymmetries of the yawed wake are explained with a difference in vortex interaction with the shear in the neutral atmospheric boundary layer.

In the test cases A and B of this study, however, neither shear or veer are present in the inflow. Nevertheless, a slight asymmetry in overall wake deflection is present, implying that other mechanisms might be the main reasons in these cases.

(2) Secondly, possible ground or side wall blockage effects are discussed. The experimental setup is perfectly symmetrical, i.e. the rotor is located in the center of the wind tunnel meaning that it has the same distance to wind tunnel floor and roof respectively the right and left sidewall. The boundary layer on floor, roof and both sidewalls is measured to be  $d_{BL,3D} \approx 20cm$  respectively  $d_{BL,6D} \approx 25cm$ . The rotor swept area blocks 12.8% of the wind tunnel cross sectional area, which affects the wake development. A LES study by Sarlak et al. (2016) showed, however, that the wake expansion is only insignificantly affected by blockage ratios smaller than 20%. For a deflected wake behind a yawed turbine, however, interactions with the sidewalls cannot be excluded anymore, especially for the higher downstream distance x/D = 6. Although the distance to each sidewall is equal, it is possible that the wake deflection is blocked to a higher degree by right sidewall (for  $\gamma = +30^{\circ}$ ) than by the left sidewall (for  $\gamma = -30^{\circ}$ ). This scenario is considered to be unlikely, however, only a high-fidelity simulation with and without wind tunnel walls could clarify this completely.

(3) As a third source for wake deflection asymmetries, systematic errors in the turbine yaw alignment should be discussed. The correct alignment at  $\gamma = 0^{\circ}$  is ensured by installing horizontal laser sheets at the central points of the wind tunnel and adjusting the turbine yaw angle to it. The yaw angle itself is adjusted with a calibrated fully automatic turntable. Inaccuracies in the experimental setup can never be excluded, however, the accuracy of the yaw angle adjustment was estimated to be within  $\pm 1^{\circ}$ . Experiments with the model turbine by ForWind as reported in the companion paper by Schottler et al. (2018) show a very symmetric wake deflection with respect to positive and negative yaw angles in an otherwise identical setup. This indicates that the slight differences in wake deflection have to be dependent on the turbine geometry or wall blockage.

(4) The final possible source for asymmetries to be discussed is the rotor wake's interaction with the tower wake. On the same rotor as used in this study, Pierella and Sætran (2017) showed that the presence of the tower wake induced significant non-symmetries in the rotor wake caused by "a different cross-stream momentum transport in the toptip and bottom-tip region." For a non-yawed turbine operated at its optimum tip speed ratio, they showed that the center of the wake vortex is slightly deflected downwards and to the left with increasing downstream distance. They are able to clearly attribute this effect to the interaction with the tower wake. As counter-evidence they managed the wake to recover its symmetric structure by installing a second mirrored turbine tower from the nacelle to the wind tunnel roof.

Pierella and Sætran's experiment indicates both a lateral and vertical displacement of the wake vortex center through the interaction with the tower wake. For the yawed case, the interaction of the counter-rotating vortex pair with the slightly displaced wake vortex might lead to a slightly differently displaced wake behind a positively and negatively yawed turbine. At this stage we only can guess about the exact interaction mechanisms, but a tower-wake-induced displaced wake vortex in the non-yawed case supports the assumption of an asymmetrically displaced wake center for the yawed cases.

In comparison to Pierella and Sætran's tower wake experiment, a slimmer tower was constructed for the new yaw experiments ( $D_{tower,old} = 61mm \text{ vs } D_{tower,new} = 43mm$ ) in order to minimize tower wake effects and adjust the geometrical scaling to a full-scale setup. The geometrical scaling of the tower now fits very well with that of a full-scale turbine (e.g. NREL 5MW reference turbine, Jonkman et al., 2009):

 $\frac{D_{tower,exp}}{D_{rotor,exp}} = \frac{0.043m}{0.894m} \approx \frac{D_{tower,NREL-5MW-ref}}{D_{rotor,NREL-5MW-ref}} = \frac{6m}{126m}$

However, a significantly larger tower drag coefficient is expected in the small-scale experiment than for a full-scale turbine. Assuming a tower diameter of

$$D_{tower,NREL-5MW-ref} = 6m$$

for a full-scale turbine, we can calculate a Reynolds number of

 $Re_{D,tower,NREL-5MW-ref} \approx 4 \times 10^6.$

According to Schlichting (1968), this is in the transition region resulting in a drag coefficient of about

 $C_{D,tower,NREL-5MW-ref} \approx 0.3.$

In our model scale experiment, however, the tower-based Reynolds number is as low as

 $Re_{D,tower,exp} \approx 3 \times 10^4$ ,

resulting in a much higher drag coefficient of

 $C_{D,tower,exp} \approx 1.0.$

Consequently, the effect of the tower wake on the rotor wake (and thus also deflected rotor wakes) is deemed to be significantly stronger in the Re-range of model-scale experiments than in full-scale situations

We share the opinion that this line of arguments for a significant influence of the tower wake on the wake deflection is not sufficiently explained in the manuscript yet. As this is a very critical issue, we suggest to add some more lines to the explanation on p.14:

**p.14, l.5 ff:**

The wake shows a higher deflection for negative yaw angles in all inflow cases. Also the wake behind the non-yawed turbine is seen to be slightly deflected in positive zdirection, which is assumed to stem from the interaction of the rotating wake with the turbine tower. As discussed by Pierella and Sætran (2017) who performed experiments on the same rotor with a slightly larger tower, the tower-wake interaction can lead leads to an uneven momentum entrainment in the wake. For the non-yawed case Pierella and Sætran (2017) observed both a lateral and vertical displacement of the wake vortex center, induced by an interaction with the tower wake. It can therefore be assumed that also the interaction of the counter-rotating vortex pair with the tower wake slightly displaced wake vortex in the yawed cases might be influence by an interaction with the tower wake, which is the only source of asymmetry in an otherwise perfectly symmetrical setup.

**Overall comment (1c)**

A final point on this discussion, could you include some discussion of the proximity of the rotor to the ceiling and the floor? I was thinking a source of discrepancy might be that LES/field data will have only a ground, and as a result only one of the vortices experiences ground effects. Is this a consideration?

This is indeed a very good thought. When discussing ground effects two different phenomena can be referred to:

(1) the presence of the ground in an otherwise uniform flow

(2) the formation of a boundary layer shear through ground friction

(1) The influence of ground effects on the interaction of a counter-rotating vortex pair (CVP) in the wake for an Actuator disc exposed to a uniform inflow has been discussed in a computational free-wake vortex filament study by Berdowski et al. (2018). In this study, ground effects could be isolated by running two different simulations, of which only one was including a symmetry plane on the ground. For this case they observed that the bottom vortex of the CVP forms another CVP with its mirror vortex underground and in opposite direction. (Berdowski et al., 2018)

As shown in Fig. 6 (c) in the manuscript, we did not observe this effect in our perfectly symmetrical experimental setup, in which both the ground and also the roof of a wind tunnel are present. Our model turbine  $(D \approx 90cm)$  is installed with a hub height  $(h_{hub,exp} = 89cm)$  adjusted to the center of the wind tunnel  $(h_{tunnel} \approx 180cm)$ . That means that about half a rotor diameter (45cm) of space is left for the freestream flow

above and below the rotor. The proximity of the rotor to the floor roughly scales with that of a full-scale turbine  $(h_{hub,NREL-5MW-ref} = 90m)$ . However, the same proximity to the ceiling is unrealistic, but was chosen to specifically to ensure the best possible symmetry in the setup and to avoid interactions with the wind tunnel boundary layers  $(d_{BL} \approx 20 - 25cm)$ . Outside of these boundary layers the inflow is spatially uniform within  $\pm 0.8\%$  (Inflow A) and  $\pm 2.5\%$  (Inflow B).

(2) In contrast to most field data and also the referenced LES simulations, where a certain amount of shear (and sometimes also veer) is present, the inflow in the wind tunnel experiment is completely uniform (Inflows A and B). That means that apart from the previously discussed tower wake effects, the interaction of the different wake vortices should be "clean" and not biased by shear or veer in the inflow. However, one could argue that the two vortices of the counter-rotating vortex pair could expand differently in a full-scale situation as the expansion of the lower vortex is limited by the ground while the upper one can expand freely. The blocked expansion of the wake and its single structures is definitely an issue in wind tunnel experiments, which becomes more serious for increasing downstream distances. It cannot be excluded that the dimension and strength of the single vortices is also influenced by wall effects in this experiment. However, comparisons of the general wake structures with experiments behind smaller, unblocked rotors show a good agreement as shown in Schottler et al. (2018) and Bartl et al. (2018). In general, it must be kept in mind that the results of this wind tunnel campaign do not reflect realistic conditions at all. A number of discrepancies as the simplifications in the inflow and especially the wall blockage can be considered as strong disadvantages to full-scale measurements and simulations. However, the controlled boundary conditions of a wind tunnel experiment allow to isolate the influence of certain parameters, i.e. inflow shear and turbulence, in a controlled manner. This can be an advantage over full-scale measurement and additionally serve as well-defined reference data for the validation of CFD codes.

**Overall comment (2)**

A second overall comment, the authors point out that is difficult to reduce wake deflection to a single value, and can complicate interpretation of results such as Fig 8-9. Since you already employ the method of available power, I believe an interesting additional comparison between the collected data and the models would be to compare the power output of an imaginary turbine located at x/D=6 and z/D=0 (and perhaps z/D = +/-0.5). This could represent an interesting assessment of do the models correctly predict the change in power obtained through wake steering for a given arrangement.

Thank you for this very good comment. This is indeed a very good idea as we actually have performed measurements available with an offset downstream turbine operated at x/D = 3. Seven different lateral offsets of the downstream turbine z/D have been chosen ranging from z/D = [-0.50, -0.33, -0.16, 0, +0.16, +0.33, +0.50]. Power measurement have been performed for the upstream turbine yaw angles  $\gamma_{T1} = 0^{\circ}$  and  $\gamma_{T1} = 30^{\circ}$ . A comparison of the Available Power calculated from mean streamwise velocity distribution in the wake with the actually measured power coefficient  $C_{P,T2}$  of a downstream rotor traversed through the wake is presented in Fig. 1.

The comparison generally shows a good match between the measured downstream turbine power and the calculated Available Power in the wake flow for both upstream turbine yaw angles. These results show that the Available Power Method generally performs as it should for the purpose of reducing the wake deflection to a single value. However, the coarse grid of only seven z/D-positions does not enable us to validated the exact location of the calculated minimum Available Power. For the calculation of the Available Power we numerically traversed the imaginary downstream turbine through 50 different offset positions from z/D = [-0.50, +0.50] allowing a location of the wake deflection with an accuracy of  $\Delta z/D \approx 0.02$ . An experimental validation with a comparable accuracy would be extremely elaborate or require an automatic traversing mechanism of the downstream turbine.

We therefore consider the presented comparison to serve as a general demonstration, but not as a sufficient validation of the *Available Power method*. We deem this demonstration not to add specific value to the discussion of our results and therefore suggest not to include this discussion in the manuscript.

Figure 1: Comparison of the Available Power calculated from mean streamwise velocity distribution in the wake with the actually measured power of an identical downstream rotor traversed through the wake at x/D = 3 for inflow B. The Available Power in an imaginary rotor swept area A traversed through the wake is multiplied with the maximum downstream turbine power coefficient  $C_{P,T2,max} = C_{P,T1,max} = 0.467$ . Vertical dashed lines indicate z/D locations of the minimum calculated Available Power.

**Specific comment (1)**

**The introduction is well done, with a good review of the literature to date. Useful to read it summarized in this way.**

Thank you! We consider adding two new references by Fleming et al. (2017) and Berdowski et al. (2018) in the introduction of the final manuscript, as some interesting new research on this topic was published in the meanwhile.

**p.2, l.29 ff:**

The topic of utilizing yaw misalignment for improved wind farm control was thoroughly investigated by Fleming et al. (2015) and Gebraad et al. (2016). They analyzed wake mitigation strategies by using both a parametric wake model and the advanced computational fluid dynamics (CFD) tool SOWFA. A recent follow-up study by Fleming et al. (2017) focused on large-scale flow structures in the wake behind one and multiple aligned turbines and addresses a wake deflection behind a non-yawed downstream impinged by a partial wake of a yawed upstream turbine.

**p.3, l.4 ff:**

 $\overline{A}$  combined experimental and computational wake study for a larger range of downstream distances was recently reported by Howland et al. (2016). The wake behind a yawed small drag disc of D=0.03 m was analyzed, describing the formation of a curled wake shape by a counter-rotating vortex pair. The influence of wake swirl, ground effect and turbulent diffusion on the formation mechanisms of this counter-rotating vortex pair was recently systematically investigated by Berdowski et al. (2018) using a free-wake vortex filament method.

**Specific comment (2)**

The selection of y as vertical and z as cross-wise was surprising to me, although since you provide a coordinate system in Fig 4., not too confusing. But is there a reason for this? FAST and Bladed for example both have z directed upward

This is a legitimate comment. Despite the unfortunate inconsistency of the coordinate system with most other publications and computational codes, we think that it is important to be consistent with our earlier publications (e.g. Bartl and Sætran (2017), Schottler et al. (2017b), Schottler et al. (2018)). We therefore carefully define the coordinate system in a clear sketch (Fig. 4 of the manuscript) before going into the results.

**Specific comment (3)**

Page 6, cos cubed is found for power-loss function. Anecdotally, this would be high for a utility-scale turbine I believe (although it fits the theoretical value). Is this a function of the scaling? Thank you for this very good comment. It seems that a number of different values for the exponent x in the power-loss function  $P(\gamma) = P_{max} \times cos^x$  have been found for different turbines of different sizes in different studies. This issue has amongst others been discussed in a thesis by Schepers (2012) as well as a review paper on yaw aerodynamics by Micallef and Sant (2016).

While earlier wind tunnel measurements at NTNU on the same rotor by Krogstad and Adaramola (2012) also find an exponent of x = 3, "other measurements by Dahlberg and Montgomery (2005) found the exponent x to vary between 1.88 and 5.14" (re-cited from Schepers, 2012). In 2001, Schepers further investigated this with another set of wind tunnel measurements and found an exponent of x = 1.8 (Schepers, 2001), which is significantly lower than the exponent found at NTNU.

It might be guessed that the exponent x could also be dependent on wind tunnel wall blockage, as blockage ( $\sigma = 12.8\%$ ) significantly influences the power characteristics of the NTNU rotor. Measurements on a downscaled NTNU rotor ( $D_{NTNU,small} = 0.45m$ ), however, confirm a power-loss-coefficient of about x = 3 (Bartl et al., 2018).

As stated by Micallef and Sant (2016), the exponent is deemed to be dependent on the induction distribution of the rotor. Therefore, a dependency of the exponent on the specific rotor design is assumed to be the main reason for the significant variations in the different experiments. A dedicated experiment on the power's yaw-dependency for different induction settings (e.g. through additional pitch or tip speed ratio variations) could help to further clarify this issue.

**Specific comment (4)**

Fig 11: I didn't understand why for the lower plots, two different methods of fitting are used. It had the impact on me, to reemphasize the difference in value of the points, since on the right the higher points are outliers to the fit.

Thank you for this good comment. We agree that the original version of Fig.11 was confusing. We assume that the values of the dotted lines in the lower left plot of the original version Fig. 11 in the manuscript might have been misunderstood. The single points shown in this subplot were the measured values of  $k/u_{ref}^2$  for  $\gamma = 0^\circ$ . These values were then multiplied with  $cos(\gamma)^2$ , which was found to be a good first order approximation for the turbulence levels for a yawed operation (shown as chain-dotted lines in the new plot). These locations of these reduced peak turbulence values are then scaled with a  $\mu \pm \sigma_u$  approximation (derived from single Gaussian fits of the mean velocity profiles) and transferred to the lower right plot. There, the approximated values are again compared with measured values (for  $\gamma = 30^\circ$ ). The whole procedure shall demonstrate that it is possible to approximate the turbulence profile in the wake of a yawed turbine, when the turbulence profile of a non-yawed turbine and mean velocity profile behind the yawed turbine are known.

For a clearer presentation of this procedure, a new version of Fig. 11 in the manuscript (Fig. 2 in this document) is suggested, only including a single Gaussian fit of the velocity profiles. All other multiple-fitted curves are omitted. Additionally, small changes in the caption and text are suggested to also make the description clearer:

**p.17, l.18 ff:**

**Effects of yawing on Approximation for turbulent kinetic energy distributions in yaw**

The levels of peak turbulence are observed to decrease considerably when the rotor is yawed. For a direct case-to-case comparison, TKE-profiles at hub height y=0 at x/D=6are presented for  $\gamma = 0^{\circ}$  and  $\gamma = -30^{\circ}$  in the lower plots of Figure 11. For a yawed turbine, the rotor thrust reduces with approximately  $\cos^2(\gamma)$  as previously shown in Figure 3. Multiplying also the TKE levels generated by the non-yawed rotor with  $\cos^2(\gamma)$  is observed to result in a decent first order approximation of the turbulence levels behind the yawed rotor. The reduced TKE levels for  $\gamma = -30^{\circ}$  are indicated by the chain-dotted lines in the lower left plot of Figure 11. In order to also find For an approximation of the lateral deflection of the turbulence peaks for yawed rotors, another first order approximation of their location can be estimated as proposed by Schottler et al. (2018) is applied. In this approach the expected value and standard deviation of the fitted a Gaussian fit of the velocity profile behind a yawed rotor is calculated. Adding the standard deviation to the expected value  $\mu \pm \sigma_u$  gives a rough estimate of the corresponding TKE peak locations of the corresponding TKE peaks, as shown by the vertical dashedlines in Figure 11. Thus, it is possible to rescale the approximate both TKE peak locations and levels by knowing TKE and mean velocity for the now-yawed case.

---

## Author Comment (AC2) · 26 Mar 2018

**Authors' response to Referee #2:**

We would like to thank the referee for reviewing this manuscript, the valuable feedback and the very constructive comments. At this stage of the review process, we respond to the referee #2's comments and propose improvements for the final manuscript. The referee's original comments are printed in **bold** followed by the corresponding answers. Passages from the manuscript are printed in *italic writing*, in which proposed additions are indicated in blue and deleted parts in .
Thank you very much for your efforts,

Jan Bartl on behalf of all authors
* * *
**Content-related remark (1)**
**In the introduction, p1.l20, you state wake redirection techniques, which intentionally apply an uneven load distribution. Instead of an uneven loading, I would say that key to wake steering is the tilting of the thrust vector. For instance, cyclic pitching results is a large uneven loading, but marginal steering, while yaw results in a much smaller uneven loading, but a large thrust vector tilting.**

Thank you for the good comment. In our understanding a tilted thrust vector and an uneven rotor load distribution (or *uneven distribution of induction*, (Micallef and Sant, 2016)) are a direct consequence of each other. Intuitively, cyclic pitching results in an uneven (cyclic) rotor loading, which then causes an unevenly distributed thrust over the rotor plane (not tilted but varying in magnitude) and consequently a tilt or yaw moment. During yaw misalignment, the thrust vector is laterally tilted, creating a yaw moment and consequently uneven rotor loads.

This is in agreement with a description by Fleming et al (2014.):
During yaw misalignment *(...) the thrust force of the turbine is shown to act along the axis of the rotor shaft. When the wind inflow is at an angle to this direction, the thrust can be divided into components fx and fy. The component fx is parallel to the flow and slows the wind, while fy is perpendicular and applies the force that causes wake redirection. IPC creates an uneven distribution of thrust forces on the rotor blades over the course of a rotation (...). This creates a tilt or yaw moment on the turbine rotor. (...) Therefore the in-plane reaction forces of the rotor on the flow are also unbalanced resulting in the fact that the turbine applies a net force on the flow perpendicular to the thrust direction, which does cause the flow to be redirected and the wake structure to be skewed.*

We do however agree that *a tilted thrust vector* intuitively is a better description for the causes of wake redirection in the context of yaw misalignment. We therefore suggest the following small modification in the manuscript:

p.1, l.19 ff:
*These methods include the reduction of the upstream turbine's axial-induction by varying its torque or blade pitch angle (Annoni et al., 2016; Bartl and Sætran, 2016) as well as wake redirection techniques, which intentionally apply  a tilted thrust vector on the front row rotors. In Fleming et al. (2015) different wake deflection mechanisms have been discussed with respect to higher wind farm power production and rotor loads.*

**Content-related remark (2)**
**On page 11, in the subsection about the tower wake deflection, you discuss several factors that contribute to the tower shadow deflection. You mention the influence of the lateral offset between the rotor and the tower during yawing, and the effect of the CVP (Counter-rotating Vortex Pair) on the wake opposite tower wake direction. What I suspect here is that the bottom of the two counter-rotating vortices is in strong interaction with its mirror image underground (i.e. the ground effect), thereby forming another CVP, but in opposite direction to the main CVP involved in the wake steering. This could hypothetically boost the deflection of the wake shadow in opposite direction to the main wake deflection.**

Thank you for this very interesting comment. We agree that the lateral offset between the rotor midpoint (center of yaw rotation) and the tower midpoint might only play a minor role in the significant deflection of the tower wake as shown in e.g. Figure 6 (a) of the manuscript. The main contribution is deemed to stem from a strong cross flow (caused by the lower vortex of the CVP) near the ground as shown in Figure 6 (b) of the manuscript.

The formation of another CVP due to the interaction with the ground as seen in Bastankhah and Porte-Agel (2016) or Berdowski et al. (2018), is not directly observed in our experimental study. An analysis of the streamwise vorticity $\omega_x$ in Figure 6 (c) of the manuscript does not clearly show the formation of another CVP near the ground. As discussed in detail in the answers to **Content-related remark (5)** later in this document, the ground effect is deemed not to play a significant role in our wind tunnel experiment. Apart from the tower, our setup is perfectly symmetrical, featuring the same distance of the rotor to the floor and the roof of the closed wind tunnel cross-section.

For a clearer distinction of the effects of the tower wake deflection, we suggest the following small additions to the text:

p.11, l.16 ff:
*On the bottom of the wake contour plot in Figure 6 (a), the wake of the turbine tower is indicated. The tower wake is observed to be deflected in the opposite direction than the rotor wake when the turbine is yawed. The deflection of the tower wake in the opposite direction is believed to have two reasons. Firstly, the turbine tower has a slight offset from $z/D = 0$ as the center of yaw-rotation was set to the rotor midpoint and not the tower. Therefore, a minor offset from the central position is expected*

*for the tower wake.* Secondly *and more importantly*, the tower wake experiences an additional deflection in opposite direction due to an adversely directed cross-flow component outside near the wind tunnel floor *as depicted in the vector plot in*(Figure 6 (b)). This cross-flow balances the counter-rotating vortex pair above and possibly deflects the tower wake further to the side.

**Content-related remark (3)**

**On page 12, the insignificant influence of the moderately sheared inflow on the wake shape is addressed. However, this can only be stated about the shear inflow under high turbulence conditions, as that is the only case you analysed. It might be the case that shear does have a significant contribution for low ambient turbulence levels, as the inflow shear in combination with the wake shear results in a distinctively high velocity gradient near the top of the wake (as shown by many researchers), thus increasing turbulence levels there. By the way, you mention this notion in the discussion about the TKE results later on in the paper.**

Thank you for this very good comment. We do completely agree that the insignificant influence of the moderately sheared inflow on the wake only holds for the investigated highly turbulent inflow. This situation might not be very realistic, as in reality stronger vertical flow gradients are mostly present in stable atmospheric conditions featuring a low ambient turbulence level (Vollmer et al., 2016). However, it is very difficult to create a low-turbulent sheared inflow in a wind tunnel experiment with a limited wind tunnel length.

We agree that we have to be clearer about this at two passages in the text, and therefore suggest the following additions:

p.1, l.6 f:
*Exposing the rotor to non-uniform highly turbulent shear inflow changes the mean and turbulent wake characteristics only insignificantly.*

p.10, l.25 f:
*Despite the sheared inflow the wake shapes for all three yaw angles and both downstream distances are observed to be very similar to those of test case B. The normalized velocity levels as well as the inner structure of the wake are almost identical. The influence of shear is however only investigated at high inflow turbulence levels, which does not allow for any conclusions at lower inflow turbulence levels.*

**Content-related remark (4)**

**For completeness, it is important that the parameter settings for the JMC and BPA models is provided.**

Thank you for pointing this out. The parameter settings are, of course, very important for the reproducibility of the deflection calculations. The recommended default

model-parameters were used in both cases. We suggest the following additions to the manuscript:

p.13, l.2 f:
*Further, the results are compared with two different wake models by Jimenez et al. (2010) (JCM) and Bastankhah and Porte-Agel (2016) (BPA).* *The recommended default model-parameters were used in the implementation of both wake deflection models. For the JCM-model a linear wake expansion factor of $\beta = 0.125$ was applied, while $k_y = 0.022$, $k_z = 0.022$, $\alpha^* = 2.32$ and $\beta^* = 0.154$ were used in the case of the BPA-model.* *The comparisons of the wake deflections are shown in Figure 8.*

**Content-related remark (5)**

**On page 14, you note that the wake deflection of a non-yawed turbine is assumed to stem from the interaction of the rotating wake with the turbine tower. The fact that the wake of a counter-clockwise rotating turbine (thus with a clockwise rotating wake swirl) deflects in positive z direction, sounds to me as originating from the interaction between the wake swirl and the ground: the root vortex forms a CVP with its mirror image underground from the ground effect, with its deflection direction in positive z direction. This was also discussed by e.g. Fleming (2014) and BPA (2016).**

This is a very good comment directed towards the core of manuscript, namely the asymmetrical interaction of the different vortices in the yawed wake. For a detailed discussion about the causes for an asymmetrical wake deflection it is also referred to the answers to **Overall comments (1b) and (1c)** in the **Authors' response to RC1**, in which a very similar comment was addressed.

The interaction of the ground with the counter-rotating vortex pair (CVP) in the wake of a yawed turbine has been discussed by Fleming et al. (2014), Bastankhah and Porte-Agel (2016) and Berdowski et al. (2018).
The study by Fleming et al. (2014) already discusses wake asymmetries influenced by the ground effect for a non-yawed turbine. *"The wake rotates counter-clockwise in these contour planes, i.e. opposite to the clockwise rotation of the turbine rotor, and the wake is like a vortex interacting with the ground. The clockwise-rotating image wake (when considering the ground plane as an image plane in potential flow) then induces motion on the actual wake, pushing it to the right."*
By the means of theoretical potential theory study Bastankhah and Porte-Agel (2016) observe a different *"wake-centre displacement (...) in both horizontal and vertical directions (...). This is due to the fact that the wake rotation and ground effects act against each other"* for one yaw direction, while they act in the same direction for the other yaw direction.
A recent computational free-wake vortex filament study by Berdowski et al. (2018) investigated the ground effect for a yawed actuator disc. In this study, ground effects could be isolated by running two different simulations, of which only one was including a symmetry plane on the ground. For this case they observed that *"the bottom vortex of the CVP forms another CVP with its mirror vortex underground and in opposite*

*direction"* (Berdowski et al., 2018).

The experimental setup investigated in this manuscript, however, is perfectly symmetrical, i.e. the rotor is located in the center of the wind tunnel, meaning that it has the same distance to wind tunnel floor and roof respectively the right and left sidewall. Our model turbine ($D \approx 90cm$) is installed with a hub height ($h_{hub,exp} = 89cm$) adjusted to the center of the wind tunnel ($h_{tunnel} \approx 180cm$), such that the setup is almost perfectly symmetrical. As shown in Fig. 6 (c) in the manuscript, we did not observe a formation of another CVP in our experimental setup.

As explained in the **Authors' response to RC1, Comment (1b)**, the effect of the tower wake on the rotor wake is deemed to be the main influence factor introducing asymmetries to the setup. However, the tower wake in this model scale experiment is deemed to be significantly stronger in the Reynolds-number-range of model-scale experiments than in full-scale situations. As this is a very critical issue, we suggest to add some more lines to the explanation on p.14 (as suggested in the answers to RC1 already):

p.14, l.5 ff:
*The wake shows a higher deflection for negative yaw angles in all inflow cases. Also the wake behind the non-yawed turbine is seen to be slightly deflected in positive z-direction, which is assumed to stem from the interaction of the rotating wake with the turbine tower. As discussed by Pierella and Sætran (2017) who performed experiments on the same rotor with a slightly larger tower, the tower-wake interaction  leads to an uneven momentum entrainment in the wake. For the non-yawed case Pierella and Sætran (2017) observed both a lateral and vertical displacement of the wake vortex center, induced by an interaction with the tower wake. It can therefore be assumed that also the interaction of the counter-rotating vortex pair with the tower wake slightly displaced wake vortex in the yawed cases might be influence by an interaction with the tower wake, which is the only source of asymmetry in an otherwise perfectly symmetrical setup.*

**Content-related remark (6)**
**On page 14, you mention that the differences are small for the wake deflection as compared between a high and low turbulence inflow. Here it would be helpful to present results of the streamwise vorticity for both cases and for several downstream positions. Maybe the diffusion of vorticity under self-induced turbulence is already very significant for low ambient turbulence levels, which would explain why both cases are then so similar. In the end, the analysis of streamwise vorticity is key to understand, as the streamwise vorticity forms the CVP which is the driving force behind both the wake deflection and the shape deformation.**

This is a very good idea for a deeper analysis. We agree the diffusion of vorticity in a field of rotor-generated turbulence for low inflow turbulence levels might be very

similar to that of higher inflow turbulence levels. However, a more detailed analysis would be needed to support this assumption.

Unfortunately, we are not able present and analyze the streamwise vorticity for all wake scans at this stage, as our Laser-Doppler-Anemometer (LDA) only allowed recording two velocity components at a time. We decided to record the streamwise component $u$ and the vertical component $v$. For an assessment of the streamwise vorticity $\omega_x$, also the lateral velocity component $w$ would be needed. This component was additionally measured for one wake scan only, which included in the parameters presented in Figure 6 of the manuscript.

As the vorticity is deemed to be of major interest for an assessment of the different diffusion in the flow, we suggest to add a line one that issue in the discussion section of the manuscript, motivating a deeper analysis of this in future studies.

p.18, l.19:

*Our study moreover indicates that the wake shape and deflection is affected by inflow turbulence.* *The overall wake deflection was observed to be similar for both investigated turbulence levels. For a more detailed investigation of diffusion mechanisms in the wake, however, a vorticity analysis in the wake of a turbine exposed to low and high turbulence is motivated for future studies. The inflow turbulence is furthermore implemented* * as an input parameter in the recently developed wake model by Bastankhah and Porte-Agel (2016).*

**Content-related remark (7)**

**In figure 11, vertical lines of the standard deviation are given, but it is unclear how the mean and standard deviation are defined here. After all, the Gaussian fit curves applied here are clearly not symmetric, thus I assume those are from a fit with multiple Gaussians, for which it is less trivial to define a mean and standard deviation. Apart from that, there is a lot of information in this figure, and it took me a while to comprehend it fully.**

Thank you for this good comment. We agree that Figure 11 of the manuscript can be misunderstood and needs to be simplified. The mean and the standard deviation are defined from a single Gaussian fit function of the mean velocity profile at hub height. Confusingly, this single Gaussian fit was not shown in the original version of Figure 11 of the manuscript. The original version included a couple of multiple Gaussian fits for the mean velocity and TKE profiles, which might have been misleading. For a clearer presentation, a new version of Figure 11 is suggested including the single Gaussian fit of the velocity profiles, while all other multiple-fitted curves are omitted. A suggested modified version of the manuscript's Fig. 11 is shown in Figure 2 of this answer document.

Additionally, small changes in the caption and text are suggested to also make the description clearer:

p.17, l.18 ff:

**Approximation for turbulent kinetic energy distributions in yaw**

The levels of peak turbulence are observed to decrease considerably when the rotor is yawed. For a direct case-to-case comparison, TKE-profiles at hub height $y=0$ at $x/D=6$ are presented for $\gamma = 0°$ and $\gamma = -30°$ in the lower plots of Figure 11.

For a yawed turbine, the rotor thrust reduces with approximately $\cos^2(\gamma)$ as previously shown in Figure 3. Multiplying also the TKE levels generated by the non-yawed rotor with $\cos^2(\gamma)$ is observed to result in a decent first order approximation of the turbulence levels behind the yawed rotor. The reduced TKE levels for $\gamma = -30°$ are indicated by the chain-dotted lines in the lower left plot of Figure 11. For an approximation of the lateral deflection of the turbulence peaks for yawed rotors, their location can be estimated as proposed by Schottler et al. (2018) . In this approach the expected value and standard deviation of  a Gaussian fit of the velocity profile behind a yawed rotor is calculated. Adding the standard deviation to the expected value $\mu \pm \sigma_u$ gives a rough estimate of the locations of the corresponding TKE peaks, as shown by the vertical lines in Figure 11. Thus, it is possible to  approximate both TKE peak locations and levels by knowing TKE and mean velocity for the now-yawed case.

[Figure]

Figure 1: **Suggested simplified version of Figure 11:** Normalized mean velocity and turbulent kinetic energy $k/u_{ref}^2$ profiles at hub height $y = 0$ and $x/D=6$. The yaw angles are set to $\gamma = 0°$ and $\gamma = -30°$. Vertical lines indicate the borders of standard deviations of Gaussian-fitted velocity profiles $\mu \pm \sigma_u$. Chain-dotted lines indicate a TKE profiles at $\gamma = 0°$ multiplied by $\cos^2(-30°)$. Dashed lines in the lower right subplot have the same magnitude as the chain-dotted lines, but are linearly scaled in $z$ to fit the peak locations of $\mu \pm \sigma_u$.

**Technical remark (1)**

**P3.l4 – "donstream" → "downstream"**
**P3.l24 – remove "used in"**
**P3.l27 – "a NREL" → "an NREL"**
**Table 1 – add full stop**
**Figure 2 – add full stop**
**Table 2 – "$CT$" → "$C_T$"**
**Table 2 – add full stop**
**P7.l3 – "a HBM" → "an HBM"**
**P8.l14 – "a eight" → "an eight"**
**P8.l16 – remove "these" before "all these"**
**P8.l16 – remove "as a result" (duplication with "is obtained")**
**P9.l22 – "an solid" → "a solid"**
**P11.l10 – add "been" in between "previously investigated"**
**P12.l13 – remove "in"**
**P13.l9 – add "the" before "BPA-model"**
**P13.l9 – lowercase for "Available"**
**P13.l16 – duplicate of the word "complex"**
**P13.l20 – "a input" → "an input"**
**P15.l13 – remove "and"**
**Figure 10 – add full stop**
**Figure 11 – remove "a" before "TKE profiles"**
**P18.l16 – "slight" → "slightly"**
**P19.l2 – "shown" → "show"**

Thank you for indicating these technical errors. We corrected all of them in the revised version of the manuscript.

**Technical remark (2)**

**Comma's could be used more extensively to increase readability. For instance, see the first paragraph of section 4.1: "At the top, the"; "As the rotor thrust is reduced, a"; "For a yawed rotor, a"; "Due to this lateral force component, the"; "Comparing the wake contours [: : :] , an asymmetry": : :**

Thank you for pointing this out. Commas have been added at the suggested passages in the text. Special attention will be given to commas in a final proof-reading of the manuscript.

**Technical remark (3)**

**Sometimes it would make the text more easily readable if the text would be broken up into several paragraphs. For instance, p13.l18: "Secondly, the wake: : :" is a confusing construction, as there is no "firstly" defined in your text. Moreover, this sentence refers to a new comparison, so to clarify the text it would be better to break it up into two sections.**

This is indeed an incorrect use of the word "Secondly,...". We replaced it with "Further,..." in the revised version of the manuscript.

**Technical remark (4)**
**At p2.l18, "The measured circulation in the wake showed clear asymmetries for positive and negative yaw angles". This is about the asymmetry of the wake regarding the kidney shape, but this sentence could also be read as an asymmetry between the values for positive and negative yaw (i.e. yaw dependency).**

We agree that the wording of the addressed sentence is equivocal. We therefore suggest a clearer wording:

p.2, l.17 ff:
*In a follow-up study, Grant and Parkin (2000) presented phase-locked particle image velocimetry (PIV) measurements in the wake. The measured circulation in the wake showed clear asymmetries in the wake shape for positive and negative yaw angles.*

**Technical remark (5)**
**In figure 1, a clockwise rotating turbine is presented in the left subfigure, while the other two subfigures depict an anti-clockwise rotating turbine. Although it was clearly mentioned in the text that the results were for a turbine that is anti-clockwise rotating, it was a bit confusing for me at first to see the picture for the clockwise rotating turbine (which I assume is the second turbine used for the experiments later on in the paper).**

Well observed. The rotor depicted in Figure 1.(a) of the manuscript turns clockwise and therefore is wrong in this setting. We updated the figure with a counter-clockwise turning rotor as shown in Figure 2, which now should be correct.

[Figure]

Figure 2: Yawed model wind turbine exposed to different inflow conditions: **(a)** $TI_A = 0.23\%$, uniform **(b)** $TI_B = 10.0\%$, uniform **(c)** $TI_C = 10.0\%$, non-uniform shear.

**Technical remark (6)**

**At page 6, it would be good to add the approximate values for cos2(30) and cos3(30) in the main text to get a feeling for their magnitude (0.75 and 0.65 respectively).**

That is a good idea. Although the approximated and actually measured values already are compared in Figure 3 of the manuscript, the approximated values are never mentioned in the text. For a value-to-value comparison with the measured $C_{P/T,\gamma=30}$ as presented in Table 2 of the manuscript, we propose the following small additions to the text:

p.6, l.12 ff:
*An approximation of this reduction can be obtained with sufficient accuracy by multiplying the maximum power of the non-yawed turbine by $C_{P,A} \cdot cos^3(30°) \approx 0.304$. An adequate estimate of the thrust coefficient of the yawed rotor can be obtained assuming a reduction by $C_{T,A} \cdot cos^2(30°) \approx 0.670$ on the thrust of the non-yawed rotor. This corresponds well to previous measurements by Krogstad and Adaramola (2012).*

**Technical remark (7)**

**In figure 5 you apply a very fine gradient scaling with contour lines added, but it is hard to extract the true magnitude from these plots. You might change these plots to one where you have much fewer gradient colors (let's say about 10), and change the colorbar accordingly (which is completely smooth in the current visualization).**

Thank you for this good comment. Matlab offers a number of different pre-defined and the option for custom defined colormaps. The most commonly used pre-defined maps are "jet" and "parula". "Jet" offers a wider spectrum of colors, which makes it easier to extract the magnitude of the values from a plot. "Parula", on the other hand, appears more linear to the eye. We therefore decided to plot our wake results using the "parula" colormap.

We deem it is very important that the colormaps are consistent with our earlier publications (e.g. Bartl et al. (2017), Schottler et al. (2018)) and therefore propose to keep the colormaps as they are in the manuscript. However, we now made all our experimental wake data publicly available on a web-platform including a digital object identifier (doi). This enables everyone to download the wake data and adjust the colormaps according to their specific preferences. We propose to add a short line called *"Data availability"* in the end of the manuscript:

p.19, l.26:
*Data availability. All presented wake data in this paper is available on https://doi.org/10.5281/zenodo.1193656 .*

**Technical remark (8)**

**In section 4.1, the subsection about the curled wake shape, you mention ": : : a kidneyshaped velocity deficit is observed: : :", without referring to a figure number. The same applies for the subsection about the tower wake deflection on the next page.**

Thank you for pointing this out. We suggest to add a reference to the specific figures in both cases:

p.10, l.32:
*At x/D=6 a kidney-shaped velocity deficit is observed (Figure 5), showing a higher local velocities behind the rotor center.*

p.11, l.16:
*On the bottom of the wake contour plots in Figure 6 (a), the wake of the turbine tower is indicated.*

**Technical remark (9)**

**In general, it comes more natural for the understanding of the reader if the lateral direction was defined as y and the vertical direction as z instead.**

This is a legitimate comment, which has also been addressed by referee 1. Despite the unfortunate inconsistency of our coordinate system with most other definitions, we think that it is important to be consistent with our earlier publications (e.g. Bartl and Sætran (2017), Schottler et al. (2017), Schottler et al. (2018)). We therefore carefully define the coordinate system in a clear sketch (Fig. 4 of the manuscript) before going into the results.

**References**

[1] Micallef, D. and Sant, T.: A Review of Wind Turbine Yaw Aerodynamics, Intech, doi: 10.5772/63445, 2016.

[2] Fleming, P., Gebraad, P. M., Lee, S., van Wingerden, J.-W., Johnson, K., Churchfield, M., Michalakes, J., Spalart, P., and Moriarty, P.: Evaluating techniques for redirecting turbine wakes using SOWFA, Renewable Energy, 70, 211–218, doi: 10.1016/j.renene.2014.02.015, 2014.

[3] Bastankhah, M. and Porté-Agel, F.: Experimental and theoretical study of wind turbine wakes in yawed conditions, Journal of Fluid Mechanics, 806, 506–541, doi: 10.1017/jfm.2016.595, 2016.

[4] Berdowski, T., Ferreira, C. van Zuijlen, A. and van Bussel, G.: Three-Dimensional Free-Wake Vortex Simulations of an Actuator Disc in Yaw, AIAA SciTech Forum, Wind Energy Synopsium 2018, doi: 10.2514/6.2018-0513, 2018.

[5] Pierella, F. and Sætran, L.: Wind tunnel investigation on the effect of the turbine tower on wind turbines wake symmetry, Wind Energy, 20, 1753-–1769, doi: 10.1002/we.2120, 2017.

[6] Vollmer, L., Steinfeld, G., Heinemann, D., and Kühn, M.: Estimating the wake deflection downstream of a wind turbine in different atmospheric stabilities: an LES study, Wind Energy Science, 1, 129–141, doi: 10.5194/wes-1-129-2016, 2016.

[7] Schottler, J., Bartl, J., Mühle, F., Sætran, L., Peinke, J., and Hölling, M.: Wind tunnel experiments on wind turbine wakes in yaw: Redefining the wake width, Wind Energy Science Discussions, doi: 10.5194/wes-2017-58, 2018.

[8] Bartl, J. Ostovan, Y., Uzol, O. and Sætran, L.: Experimental study on power curtailment of three in-line turbines, Energy Procedia, vol. 137C, pp. 307-314, doi: 10.1016/j.egypro.2017.10.355, 2017.

[9] Bartl, J. and Sætran, L.: Blind test comparison of the performance and wake flow between two in-line wind turbines exposed to different turbulent inflow conditions, Wind Energ. Sci., 2, 55–76, doi: 10.5194/wes-2-55-2017, 2017.

[10] Schottler, J., Mühle, F., Bartl, J., Peinke, J., Adaramola, M.S., Sætran, L. and Hölling, M.: Comparative study on the wake deflection behind yawed wind turbine models, Journal of Physics: Conference Series, Volume 854, doi: 10.1088/1742-6596/854/1/012032, 2017b.